# Control of *Arabidopsis* shoot stem cell homeostasis by two antagonistic CLE peptide signalling pathways

Jenia Schlegel, Gregoire Denay, Rene Wink, Karine Gustavo Pinto, Yvonne Stahl, Julia Schmid, Patrick Blümke, Rüdiger GW Simon*

Institute for Developmental Genetics and Cluster of Excellence on Plant Sciences, Heinrich Heine University, Düsseldorf, Germany

**Abstract** Stem cell homeostasis in plant shoot meristems requires tight coordination between stem cell proliferation and cell differentiation. In *Arabidopsis*, stem cells express the secreted dodecapeptide CLAVATA3 (CLV3), which signals through the leucine-rich repeat (LRR)-receptor kinase CLAVATA1 (CLV1) and related CLV1-family members to downregulate expression of the homeodomain transcription factor *WUSCHEL* (*WUS*). WUS protein moves from cells below the stem cell domain to the meristem tip and promotes stem cell identity, together with *CLV3* expression, generating a negative feedback loop. How stem cell activity in the meristem centre is coordinated with organ initiation and cell differentiation at the periphery is unknown. We show here that the *CLE40* gene, encoding a secreted peptide closely related to CLV3, is expressed in the SAM in differentiating cells in a pattern complementary to that of *CLV3*. *CLE40* promotes *WUS* expression via BAM1, a CLV1-family receptor, and *CLE40* expression is in turn repressed in a *WUS*-dependent manner. Together, *CLE40-BAM1-WUS* establish a second negative feedback loop. We propose that stem cell homeostasis is achieved through two intertwined pathways that adjust WUS activity and incorporate information on the size of the stem cell domain, via *CLV3-CLV1*, and on cell differentiation via *CLE40-BAM1*.

*For correspondence:
ruediger.simon@hhu.de

Competing interest: The authors declare that no competing interests exist.

## Introduction

In angiosperms, the stem cell domain in shoot meristem is controlled by the directional interplay of two adjacent groups of cells. These are the central zone (CZ) at the tip of the dome-shaped meristem, comprising slowly dividing stem cells, and the underlying cells of the organizing centre (OC). Upon stem cell division, daughter cells are displaced laterally into the peripheral zone (PZ), where they can enter differentiation pathways (*Fletcher et al., 1999*; *Hall and Watt, 1989*; *Reddy et al., 2004*; *Schnablová et al., 2020*; *Stahl and Simon, 2005*; *Steeves and Sussex, 1989*). Cells in the OC express the homeodomain transcription factor WUSCHEL (WUS), which moves through plasmodesmata to CZ cells to maintain stem cell fate and promote expression of the secreted signalling peptide CLAVATA3 (*CLV3*) (*Brand et al., 2000*; *Daum et al., 2014*; *Müller et al., 2006*; *Schoof et al., 2000*; *Yadav et al., 2011*). Perception of CLV3 by plasma membrane-localized receptors in the OC cells triggers a signal transduction cascade and downregulates WUS activity, thus establishing a negative feedback loop (*Mayer et al., 1998*; *Ogawa et al., 2008*; *Yadav et al., 2011*). Mutants of *CLV3* or its receptors (see below) fail to confine *WUS* expression and cause stem cell proliferation, while *WUS* mutants cannot maintain an active stem cell population (*Brand et al., 2002*; *Clark et al., 1993*; *Clark et al., 1995*; *Endrizzi et al., 1996*; *Laux et al., 1996*; *Schoof et al., 2000*). WUS function in the OC is negatively regulated by HAM transcription factors, and only WUS protein that moves upwards to the stem cell zone, which lacks *HAM* expression, can activate *CLV3* expression (*Han et al., 2020*; *Zhou*

**eLife digest** Plants are sessile lifeforms that have evolved many ways to overcome this challenge. For example, they can quickly adapt to their environment, and they can grow new organs, such as leaves and flowers, throughout their lifetime.

Stem cells are important precursor cells in plants (and animals) that can divide and specialize into other types of cells to help regrow leaves and flowers. A region in the plant called meristem, which can be found in the roots and shoots, continuously produces new organs in the peripheral zone of the meristem by maintaining a small group of stem cells in the central zone of the meristem.

This is regulated by a signalling pathway called CLV and a molecule produced by the stem cells in the central zone, called CLV3. Together, they keep a protein called WUS (found in the deeper meristem known as the organizing zone) at low levels. WUS, in turn, increases the production of stem cells that generate CLV3. However, so far it was unclear how the number of stem cells is coordinated with the rate of organ production in the peripheral zone.

To find out more, Schlegel et al. studied cells in the shoot meristems from the thale cress *Arabidopsis thaliana*. The researchers found that cells in the peripheral zone produce a molecule called CLE40, which is similar to CLV3. Unlike CLV3, however, CLE40 boosts the levels of WUS, thereby increasing the number of stem cells. In return, WUS reduces the production of CLE40 in the central zone and the organizing centre. This system allows meristems to adapt to growing at different speeds.

These results help reveal how the activity of plant meristems is regulated to enable plants to grow new structures throughout their life. Together, CLV3 and CLE40 signalling in meristems regulate stem cells to maintain a small population that is able to respond to changing growth rates. This understanding of stem cell control could be further developed to improve the productivity of crops.

*et al., 2018*). The *CLV3-WUS* interaction can serve to maintain the relative sizes of the CZ and OC, and thereby meristem growth along the apical-basal axis. However, cell loss from the PZ due to production of lateral organs requires a compensatory size increase of the stem cell domain.

The CLV3 signalling pathway, which acts along the apical-basal axis of the meristem, has been widely studied in several plant species and shown to be crucial for stem cell homeostasis in shoot and floral meristems (*Somssich et al., 2016*). The CLV3 peptide is perceived by a leucine-rich repeat (LRR) receptor kinase, CLAVATA1 (CLV1), which interacts with coreceptors of the CLAVATA3 INSENSITIVE RECEPTOR KINASES (CIK) 1–4 family (*Clark et al., 1997*; *Cui et al., 2018*). CLV1 activation involves autophoshorylation, interaction with membrane-associated and cytosolic kinases and phosphatases (*Blümke et al., 2021*; *Defalco et al., 2021*). Furthermore, heterotrimeric G-proteins and MAPKs have been implicated in this signal transduction cascade in maize and *Arabidopsis* (*Betsuyaku et al., 2011*; *Bommert et al., 2013*; *Ishida et al., 2014*; *Lee et al., 2019*). Besides CLV1, several other receptors contribute to WUS regulation, among them RECEPTOR-LIKE PROTEIN KINASE2 (RPK2), the CLAVATA2-CORYNE heteromer (CLV2-CRN) and BARELY ANY MERISTEM1-3 (BAM1-3) (*Bleckmann et al., 2010*; *Deyoung and Clark, 2008*; *Hord et al., 2006*; *Jeong et al., 1999*; *Kinoshita et al., 2010*; *Müller et al., 2008*). The BAM receptors share high-sequence similarity with CLV1 and perform diverse functions throughout plant development. Double mutants of *BAM1* and *BAM2* maintain smaller shoot and floral meristems, thus displaying the opposite phenotype to mutants of *CLV1* (*DeYoung et al., 2006*; *Deyoung and Clark, 2008*; *Hord et al., 2006*). Interestingly, ectopic expression experiments showed that CLV1 and BAM1 can perform similar functions in stem cell control (*Nimchuk et al., 2015*). In addition, one study showed that CLV3 could interact with CLV1 and BAM1 in cell extracts (*Shinohara and Matsubayashi, 2015*), although another in vitro study did not detect *BAM1-CLV3* interaction at physiological levels of CLV3 (*Crook et al., 2020*). Furthermore, CLV1 was shown to act as a negative regulator of *BAM1* expression, which was interpreted as a genetic buffering system, whereby a loss of CLV1 is compensated by upregulation of BAM1 in the meristem centre (*Nimchuk, 2017*; *Nimchuk et al., 2015*). Comparable genetic compensation models for CLE peptide signalling in stem cell homeostasis were established for other species, such as tomato and maize (*Rodriguez-Leal et al., 2019*).

Maintaining the overall architecture of the shoot apical meristem during the entire life cycle of the plant requires replenishment of differentiating stem cell descendants in the PZ, indicating that

cell division rates and cell fate changes in both regions are closely connected (*Stahl and Simon, 2005*). Overall meristem size is restricted by the ERECTA-family signalling pathway, which is activated by EPIDERMAL PATTERNING FACTOR (EPF)-LIKE (EPFL) ligands from the meristem periphery and confines both *CLV3* and *WUS* expression (*Mandel et al., 2014*; *Shpak, 2013*; *Shpak et al., 2004*; *Torii et al., 1996*; *Zhang et al., 2021*). In the land plant lineage, the shoot meristems of bryophytes such as the moss *Physcomitrium patens* appear less complex than those of angiosperms and carry only a single apical stem cell which ensures organ initiation by continuous asymmetric cell divisions (*de Keijzer et al., 2021*; *Harrison et al., 2009*). Broadly expressed CLE peptides were here found to restrict stem cell identity and act in division plane control (*Whitewoods et al., 2018*). Proliferation of the apical notch cell in the liverwort *Marchantia polymorpha* is promoted by MpCLE2 peptide which acts from outside the stem cell domain via the receptor MpCLV1, while cell proliferation is confined by MpCLE1 peptide through a different receptor (*Hata and Kyozuka, 2021*; *Hirakawa et al., 2019*; *Hirakawa et al., 2020*; *Takahashi et al., 2021*). Thus, antagonistic control of stem cell activities through diverse CLE peptides is conserved between distantly related land plants. In the grasses, several CLEs were found to control the stem cell domain. In maize, *ZmCLE7* is expressed from the meristem tip, while *ZmFCP1* is expressed in the meristem periphery and its centre. Both peptides restrict stem cell fate via independent receptor signalling pathways (*Liu et al., 2021*; *Rodriguez-Leal et al., 2019*). In rice, overexpression of the CLE peptides OsFCP1 and OsFCP2 downregulates the homeobox gene *OSH1* and arrests meristem function (*Ohmori et al., 2013*; *Suzaki et al., 2008*). Common for rice and maize, CLE peptide signalling restricts stem cell activities in the shoot meristem, but a stem cell-promoting pathway was not identified so far.

Importantly, how stem cell activities in the CZ and OC are coordinated to regulate organ initiation and cell differentiation in the PZ, which is crucial to maintain an active meristem, is not yet known. In maize, the CLV3-related peptide ZmFCP1 was suggested to be expressed in primordia and convey a repressive signal on the stem cell domain (*Je et al., 2016*). In *Arabidopsis*, the most closely related peptide to CLV3 is CLE40, which was shown to act in the root meristem to restrict columella stem cell fate and regulate the expression of the *WUS* paralog *WOX5* (*Berckmans et al., 2020*; *Hobe et al., 2003*; *Pallakies and Simon, 2014*; *Stahl et al., 2013*; *Stahl and Simon, 2010*). Endogenous functions of CLE40 in the SAM have not previously been described, although overexpression of *CLE40* causes shoot stem cell termination, while *CLE40* expression from the *CLV3* promoter fully complements the shoot and floral meristem defects of *clv3* mutants (*Hobe et al., 2003*). We therefore hypothesized that *CLE40* could act in a *CLV3*-related pathway in shoot stem cell control.

Here, we show that the expression level of *WUS* in the OC is subject to feedback regulation from the PZ, which is mediated by the secreted peptide CLE40. In the shoot meristem, *CLE40* is expressed in a complementary pattern to *CLV3* and excluded from the CZ and OC. In *cle40* loss-of-function mutants, *WUS* expression is reduced, and shoot meristems remain small and flat, indicating that *CLE40* signalling is required to maintain *WUS* expression in the OC. Ectopic expression of *WUS* represses *CLE40* expression, while in *wus* loss-of-function mutants *CLE40* is expressed in the meristem centre, indicating that CLE40, in contrast to CLV3, is subject to negative feedback regulation by WUS. CLE40 likely acts as an autocrine signal that is perceived by BAM1 in a domain flanking the OC.

Based on our findings, we propose a new model for the regulation of the stem cell domain in the shoot meristem in which signals and information from both the CZ and the PZ are integrated through two interconnected negative feedback loops that sculpt the dome-shaped shoot meristems of angiosperms.

## Results

### CLE40 signalling promotes IFM growth from the PZ

Previous studies showed that *CLE40* expression from the *CLV3* promoter can fully complement a *clv3-2* mutant, indicating that CLE40 can substitute CLV3 function in the shoot meristem to control stem cell homeostasis, if expressed from the stem cell domain. Furthermore, while all other *CLE* genes in *Arabidopsis* lack introns, the *CLE40* and *CLV3* genes carry two introns at very similar positions (*Hobe et al., 2003*), indicating close evolutionary relatedness. Phylogenetic analysis revealed that CLV3 and CLE40 locate in the same cluster together with CLV3 orthologues from rice, maize and tomato (*Goad et al., 2017*). The amino acid sequences of CLV3 and CLE40 differ at 4 out of 13 positions (*Figure 1A*).

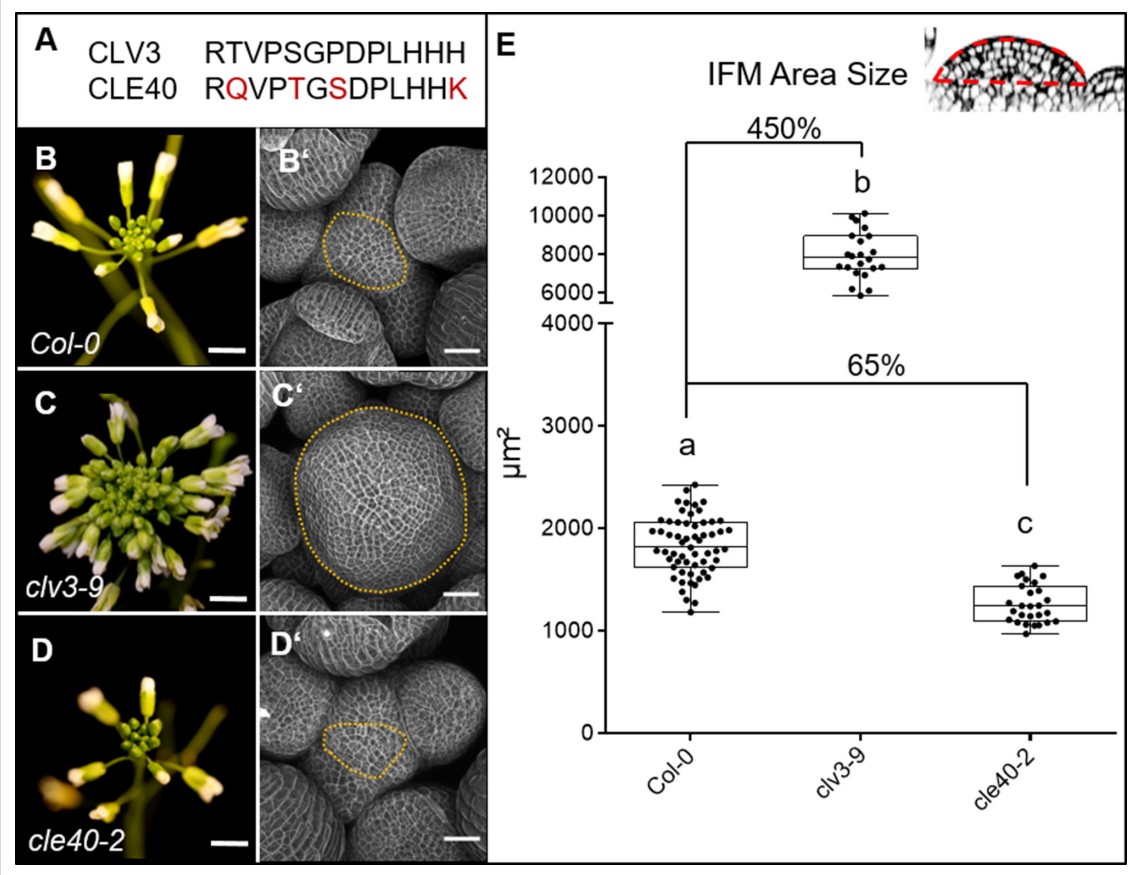

**Figure 1.** CLV3 and CLE40 exert opposite effects on meristem size. (**A**) The amino acid (AA) sequences of the mature CLV3 and CLE40 peptides differ in four AAs (differences marked in red). (**B**) *Col-0* inflorescence at 6 weeks after germination (WAG) with flowers. (**B′**) Inflorescence meristem (IFM) at 6 WAG, maximum intensity projection (MIP) of a z-stack taken by confocal microscopy. (**C**) *clv3-9* inflorescence at 6 WAG (**C′**) MIP of a *clv3-9* IFM at 6 WAG. (**D**) Inflorescence of *cle40-2* at 6 WAG (**D′**) MIP of a *cle40-2* IFM. (**E**) Box and whisker plot of IFM sizes of *Col-0* (N = 59), *clv3-9* (N = 22) and *cle40-2* (N = 27) plants. Scale bars: 10 mm (**B–D**), 50 μm (**B′–D′**), Statistical groups were assigned after calculating p-values by ANOVA and Tukey's multiple comparison test (differential grouping from p≤0.01). Yellow dotted lines in (**B′–D′**) enclose the IFM, red line in the inset meristem in (**E**) indicates the area that was used for the quantifications in (**E**).

The online version of this article includes the following figure supplement(s) for figure 1:

**Figure supplement 1.** Mutants from the CLV pathway show differences in their leaf lengths.

**Figure supplement 2.** *cle40* mutants have smaller meristems.

**Figure supplement 3.** Siliques of various mutants differ in their carpel number.

Mutations in *CLE40* were previously found to affect distal stem cell maintenance in the root meristem, revealing that a CLV3-related signalling pathway also operates in the root stem cell niche. To uncover a potential role of *CLE40* in shoot development, we analysed seedling and flower development, and inflorescence meristem (IFM) sizes of the wild-type *Col-0*, and *clv3-9* and *cle40-2 loss-of-function* mutants. At 4 weeks after germination (WAG), leaves of *clv3-9* mutants remained shorter than those of *Col-0* or *cle40-2* (*Figure 1—figure supplement 1*). After floral induction, the inflorescences of *clv3-9* mutants were compact with many more flowers than the wild type, while *cle40-2* mutant inflorescences appeared smaller than the control (B–D). To first investigate effects on meristem development in detail, longitudinal sections through the IFM at 6 WAG were obtained by confocal microscopy and meristem areas were analysed (*Figure 1B–E*). In *clv3-9* mutants, meristem areas increased to ~450% of wild-type (*Col-0*) levels, while shoot meristems from four independent *cle40* mutant alleles in a *Col-0* background (*cle40-2, cle40-cr1, cle40-cr2, cle40-cr3*) reached only up to 65% of wild type (*Figure 1E*, *Figure 1—figure supplement 2C*; *Yamaguchi et al., 2017*). Next, we used carpel number as a rough proxy for flower meristem (FM) size, which was 2 ± 0.0 (N = 290)

in *Col-0* and *cle40-2* (N = 290) but 3.7 ± 0.4 (N = 340) in *clv3-9* (**Figure 1—figure supplement 3**). Hence, we concluded that CLE40 mainly promotes IFM growth, whereas CLV3 serves to restrict both IFM and FM sizes.

We next analysed the precise *CLE40* expression pattern using a transcriptional reporter line, *CLE40:Venus-H2B*, which showed the same expression pattern as a previously described reporter (**Stahl et al., 2009**; **Figure 2—figure supplement 1**). We first concentrated on the IFMs and FMs. *CLE40* is expressed in IFMs and FMs, starting at P5 to P6 onwards (**Figure 2A–C**). We found stronger expression in the PZ than in the CZ, and no expression in young primordia. Using MorphoGraphX software, we extracted the fluorescence signal originating from the outermost cell layer (L1) of the IFM and noted downregulation of *CLE40* expression in the centre of the meristem (**Figure 2B**). Longitudinal optical sections through the IFM showed that *CLE40* is not expressed in the CZ, and only occasionally in the OC region (**Figure 2C**, **Figure 2—figure supplement 2A-E'**). Expression of *CLE40* changed dynamically during development: expression was concentrated in the IFM, but lacking at sites of primordia initiation (P0 to P4/5, **Figure 2C**). In older primordia from P5/6 onwards, *CLE40* expression is detectable from the centre of the young FM and expands towards the FM periphery. In the FMs, *CLE40* is lacking in young sepal primordia (P6), but starts to be expressed on the adaxial sides of petals at P7 (**Figure 2A**, P1–P7).

To compare the *CLE40* pattern with that of *CLV3*, we introgressed a *CLV3:NLS-3xmCherry* transcriptional reporter into the *CLE40:Venus-H2B* background. *CLV3* and *CLE40* are expressed in almost mutually exclusive domains of the IFM, with *CLV3* in the CZ surrounded by *CLE40* expressing cells (**Figure 2D–F"**, **Figure 2—figure supplement 2**). In the deeper region of the IFM, where the OC is located, both *CLV3* and *CLE40* are not expressed (**Figure 2F**, **Figure 2—figure supplement 2**).

We noted that *CLE40* is downregulated where *WUS* is expressed, or where WUS protein localizes, such as the OC and CZ. Furthermore, *CLE40* is also lacking in very early flower primordia and incipient organs.

## *CLE40* expression is repressed by WUS activity

To further analyse the regulation of *CLE40* expression, we introduced the *CLE40* transcriptional reporter into the *clv3-9* mutant background (**Figure 3A and B**, **Figure 3—figure supplement 1**). In *clv3-9* mutants, *WUS* is no longer repressed by the *CLV* signalling pathway, and the CZ of the meristem increases in size as described previously (**Clark et al., 1995**). In the *clv3-9* mutant meristems, both *CLV3* and *WUS* promoter activity is now found in an expanded domain (**Figure 3—figure supplement 1**). *CLE40* is not expressed in the tip and centre of the IFM but is rather confined to the peripheral domain, where neither *CLV3* nor *WUS* are expressed (**Figure 3B'**, **Figure 3—figure supplement 1B'**). To further explore the expression dynamics of *CLE40* in connection with regulation of stem cell fate and WUS, we misexpressed *WUS* from the *CLV3* promoter and introgressed it into plants carrying the *CLE40:Venus-H2B* construct. Since WUS activates the *CLV3* promoter, *CLV3:WUS* misexpression triggers a positive feedback loop. This results in a continuous enlargement of the CZ (**Brand et al., 2002**). Young seedlings carrying the *CLV3:WUS* transgene at 10 days after germination (DAG) displayed a drastically enlarged SAM compared to wild-type seedlings of the same age (**Figure 3C–D'**). Wild-type seedlings at this stage express *CLE40* in older leaf primordia and deeper regions of the vegetative SAM (**Figure 3E–E'**). The *CLV3:WUS* transgenic seedlings do not initiate lateral organs from the expanded meristem, and *CLE40* expression is confined to the cotyledons (**Figure 3F–F'**). *CLE40* is also lacking in the deeper regions of the vegetative SAM (**Figure 3F'**). Thus. we conclude that either WUS itself or a WUS-dependent regulatory pathway represses *CLE40* gene expression.

We next determined if *CLE40* repression in the CZ can be alleviated in mutants with reduced WUS activity. Since *wus* loss-of-function mutants fail to maintain an active CZ and shoot meristem, we used the hypomorphic *wus-7* allele (**Graf et al., 2010**; **Ma et al., 2019**). *wus-7* mutants are developmentally delayed. Furthermore, *wus-7* mutants generate an IFM, but the FMs give rise to sterile flowers that lack inner organs (**Figure 3—figure supplement 2**). We introgressed the *CLE40* reporter into *wus-7* and found that at 5 WAG all *wus-7* mutants expressed *CLE40* in both the CZ and the OC of the IFM (**Figure 3G–H'**, **Figure 3—figure supplement 2**). Similar to wild type, *CLE40* is only weakly expressed in the young primordia of *wus-7*. Therefore, we conclude that a *WUS*-dependent pathway downregulates *CLE40* in the centre of the IFM during normal development.

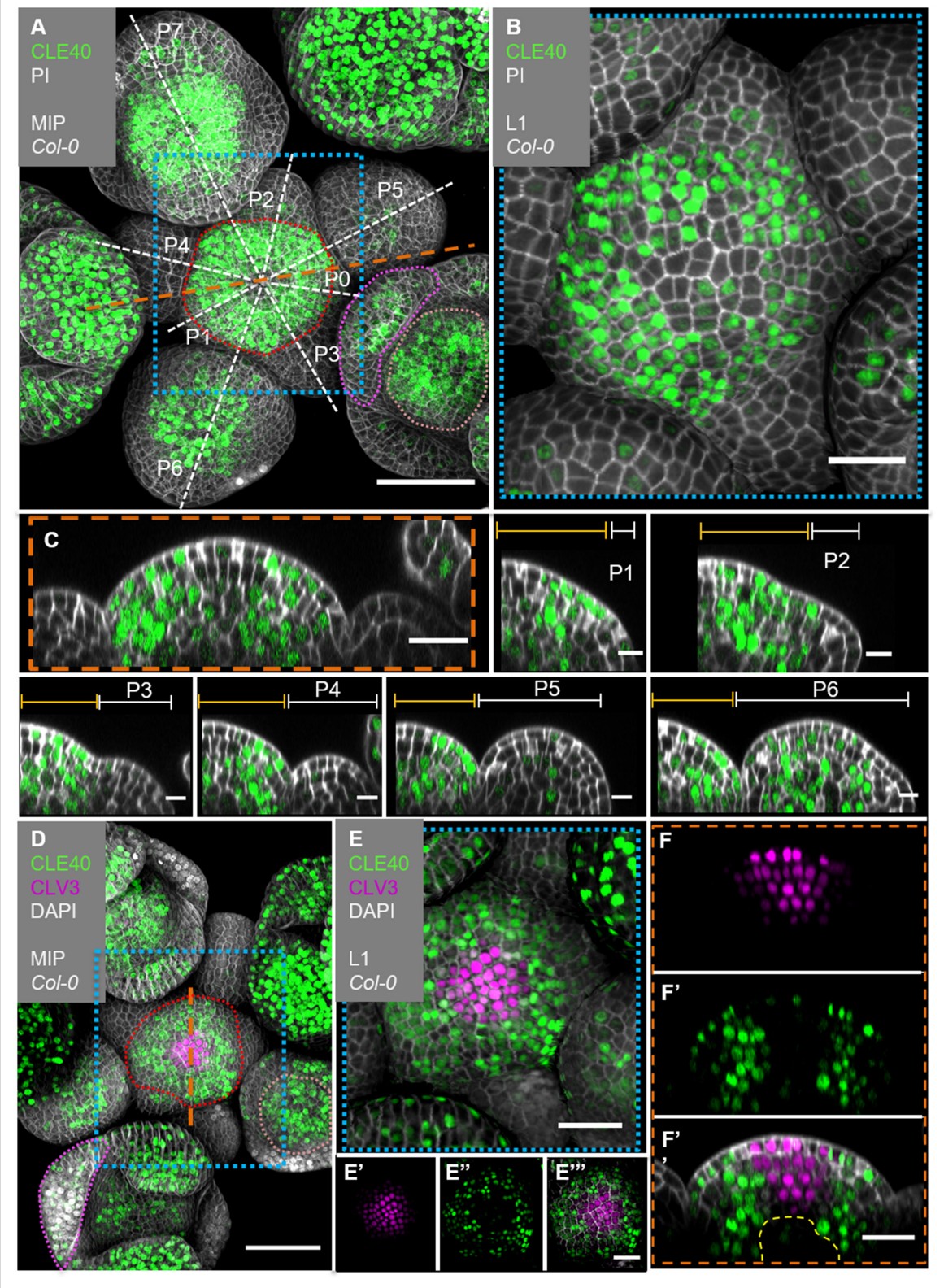

**Figure 2.** *CLE40* and *CLV3* show complementary expression patterns in the inflorescence meristem (IFM). (**A**) Maximum intensity projection (MIP) of an inflorescence at 5 weeks after germination (WAG) expressing the transcriptional reporter *CLE40:Venus-H2B//Col-0* showing *CLE40* expression in the IFM, older primordia and sepals (N = 23). (**B**) The L1 projection shows high expression in the epidermis of the periphery of the IFM and only weak expression in the central zone (CZ). (**C**) Longitudinal section through the IFM shows expression of *CLE40* in the periphery, but downregulated expression

*Figure 2 continued on next page*

*Figure 2 continued*

in the CZ. (**P1–P6**) Longitudinal section through primordia show no *CLE40* expression in young primordia (**P1–P4**), but in the centre of older primordia (**P5–P6**). (**D**) The MIP of the double reporter line of *CLE40* and *CLV3* (*CLE40:Venus-H2B;CLV3:NLS-3xmCherry//Col-0*) shows *CLV3* expression in the CZ surrounded by *CLE40* expression in the periphery (N = 12). (**E–E'''**) The L1 projection shows *CLV3* (**E'**) expression in the centre of the IFM and *CLE40* (**E''**) expression in a distinct complementary pattern in the periphery of the IFM. (**F**) The longitudinal section through the centre of the IFM shows *CLV3* expression in the CZ while *CLE40* (**F'**) is mostly expressed in the surrounding cells. (**F''**) *CLE40* and *CLV3* are expressed in complementary patterns. Dashed blue lines indicate magnified areas, dashed white and orange lines indicate planes of longitudinal sections, dashed red line in (**A**) and (**D**) marks the IFM area, the dashed pink line marks the sepals, the dashed rose line marks the FMs and dashed yellow line in (**F''**) the OC. Scale bars: 50 µm (**A, D**), 20 µm (**B, C, E, E''', F''**), 10 µm (P0–P6), PI: propidium iodide; L1: visualization of layer 1 only; P1–P7: primordia at consecutive stages.

The online version of this article includes the following figure supplement(s) for figure 2:

**Figure supplement 1.** *CLE40* transcriptional and translational reporter lines display same expression patterns.

**Figure supplement 2.** *CLE40* and *CLV3* expression in multiple inflorescence meristem (IFM).

## CLE40 signals through BAM1

Given that CLV1 and BAM1 perform partially redundant functions to perceive CLV3 in shoot and floral meristems, we asked if these receptors also contribute in a CLE40 signalling pathway. We therefore generated the translational reporter lines *CLV1:CLV1-GFP* and *BAM1:BAM1-GFP*, and analysed their expression patterns in detail. Both reporter lines rescued the shoot phenotype in a *clv1-101* and *bam1-3;clv1-20* mutant background, respectively. The *CLV1* reporter line additionally showed the same expression pattern in the shoot compared to a previously published reporter line (***Nimchuk et al., 2011***; ***Figure 4—figure supplement 1***, ***Figure 4—figure supplement 2***, ***Figure 4—figure supplement 3***, ***Figure 5—figure supplement 1***). We observed dynamic changes of *CLV1* expression during the different stages of flower primordia initiation. *CLV1:CLV1-GFP* is continuously expressed in deeper regions of the IFM comprising the OC, and in the meristem periphery where new FMs are initiating (***Figure 4A***, ***Figure 4—figure supplement 2***, ***Figure 4—figure supplement 3***). *CLV1* is expressed strongly in cells of the L1 and L2 of incipient organ primordia (P-1, P0), and only in L2 at P1. P2 and P3 show only very faint expression in the L1, but in stages from P4 to P6, *CLV1* expression expands from the L3 into the L2 and L1 (***Figure 4***, P1–P6, ***Figure 4—figure supplement 2***, ***Figure 4—figure supplement 3***).

The translational *BAM1:BAM1-GFP* reporter is expressed in the IFM, the FMs and in floral organs (***Figure 5A***, ***Figure 5—figure supplement 2A–C'***, ***Figure 5—figure supplement 3A–E'***). In the IFM, expression is found throughout the L1 layer of the meristem, and, at an elevated level, in L2 and L3 cells of the PZ, but not in the meristem centre around the OC, where *CLV1* expression is detected (***Figure 5B and C***, compare to ***Figure 4C***). *BAM1* is less expressed in the deeper regions of primordia from P6 onwards (***Figure 5C***, ***Figure 5—figure supplement 3A'–E'***). *BAM1* transcription was reported to be upregulated in the meristem centre in the absence of CLV3 or CLV1 signalling (***Nimchuk, 2017***). Using our translational BAM1 reporter in the *clv1-20* mutant background, we quantified and thereby confirmed that *BAM1* is now expressed in the meristem centre, similar to the pattern of *CLV1* in the wild type, and that *BAM1* is upregulated in the L1 of the meristem. Importantly, in a *clv1-20* background BAM1 is absent in the peripheral region of the IFM and the L2 (***Figure 5D–F***, ***Figure 4—figure supplement 3F–J'***, ***Figure 5—figure supplement 4***).

In longitudinal and transversal optical sections through the IFM, we found that complementarity of *CLE40* and *CLV3* is reflected in the complementary expression patterns of *BAM1* and *CLV1* (***Figure 5—figure supplement 5***). Therefore, we conclude that expression patterns of *CLV1* and *BAM1* are mostly complementary in the meristem itself and during primordia development. When comparing *CLE40* and *BAM1* expression patterns, we found a strong overlap in the PZ of the meristem, during incipient primordia formation, in older primordia and in L3 cells surrounding the OC (***Figure 5—figure supplement 5A' and B'***, ***Figure 5—figure supplement 6***). Similarly, CLV3 and CLV1 are confined to the CZ and OC, respectively.

To analyse if CLE40-dependent signalling requires CLV1 or BAM1, we measured the sizes of IFMs in the respective single and double mutants (***Figure 6***). While *cle40-2* mutant IFMs reached 65% of the wild-type size, *clv1-101* plants develop IFMs that were 140% wild-type size, whereas *bam1-3;clv1-101* double mutant meristems reached 450% wild-type size, similar to those of *clv3-9* mutants. This supports the notion that BAM1 can partially compensate for CLV1 function in the CLV3 signalling pathway when expressed in the meristem centre (***Figure 5F***; ***Nimchuk et al., 2015***). The relationship

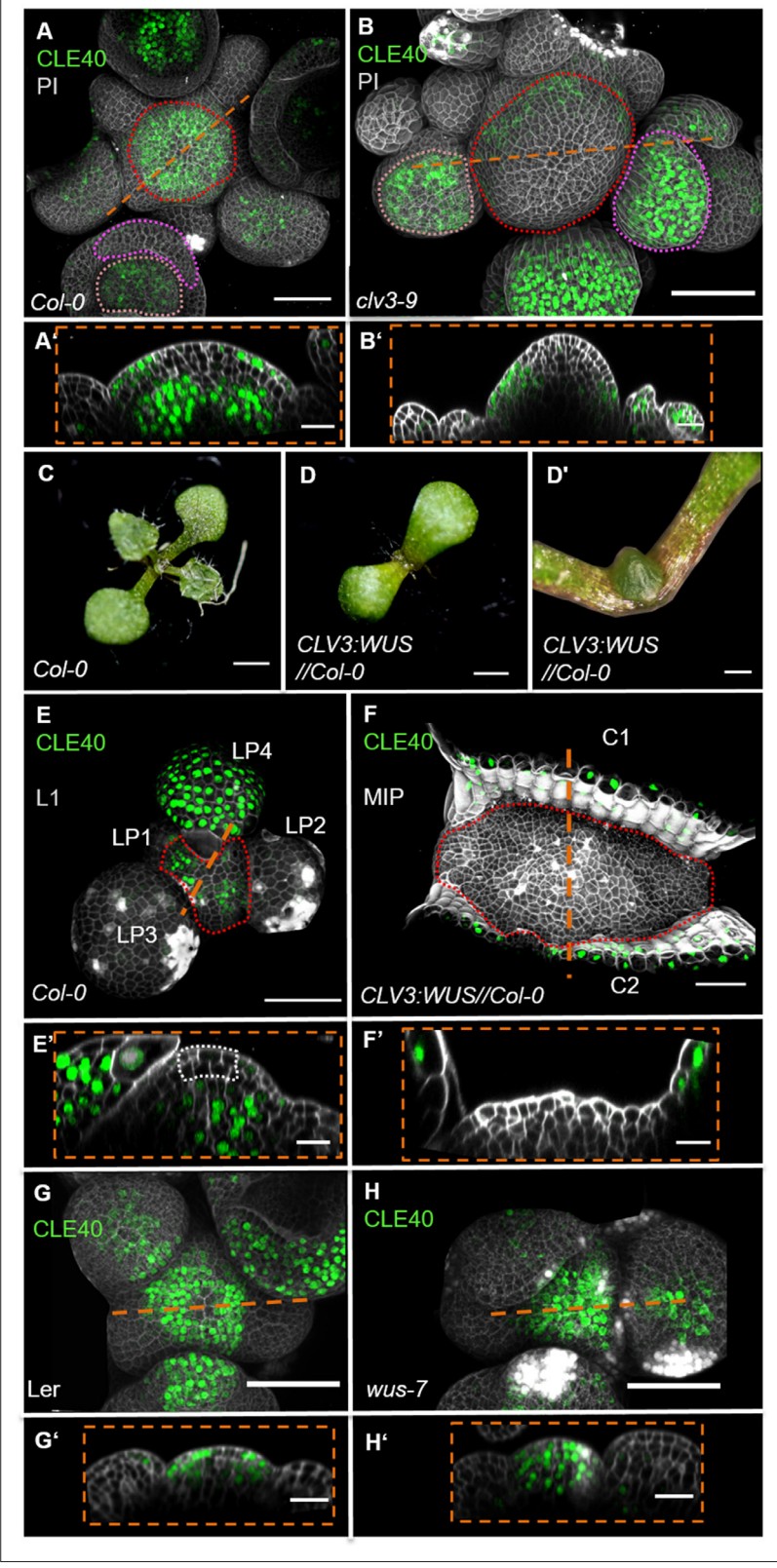

**Figure 3.** *WUS*-dependent repression of *CLE40* expression in the shoot meristem. (**A**) Maximum intensity projection (MIP) of *CLE40* expression (*CLE40:Venus-H2B//Col-0*) at 5 weeks after germination (WAG), (**A'**) Longitudinal optical section through the centre of the inflorescence meristem (IFM) (indicated by orange line in **A**) reveals no *CLE40* expression in the central zone (CZ) and the centre of the meristem. Cells in the L2 layer also

*Figure 3 continued on next page*

*Figure 3 continued*

show less *CLE40* expression. High *CLE40* expression is found in the peripheral zone (PZ) (N = 23). (**B**) MIP of *CLE40* expression in a *clv3-9* mutant (*CLE40:Venus-H2B//clv3-9*) shows expression only in the PZ of the meristem, in flower meristems (FMs) and in sepals (N = 6). (**B'**) Longitudinal optical section through the IFM depicts no *CLE40* expression at the tip and the centre of the meristem. *CLE40* expression is only detected in cells at the flanks of the IFM and in sepals. (**C**) *Arabidopsis* seedling at 10 days after germination (DAG). (**D**) Seedling expressing *WUS* from the *CLV3* promoter, 10 DAG. (**D'**) Magnification of seedling in (**D**). The meristem fasciates without forming flowers. (**E**) L1 projection, vegetative seedling with *CLE40* expression in the PZ and leaf primordia starting from LP4, at 10 DAG (N = 5). (**E'**) Longitudinal section of (**E**) with *CLE40* expression primordia and rib meristem or periphery. (**F**) MIP of fasciated meristem as in (**D**). *CLE40* expression can only be found in the cotyledons (C1 and C2) next to the meristem (N = 5). (**F'**) Longitudinal optical section shows *CLE40* expression only in the epidermis of cotyledons. (**G, G'**) MIP (**G**) and longitudinal optical section (**G'**) of *CLE40* expression (*CLE40:Venus-H2B//Ler*) in a wild- type (Landsberg *erecta* [L.*er*]) background at 5 WAG shows no signal in the CZ or OC. *CLE40* is confined to the PZ and the centre of older flower primordia, and to sepals (N = 8). (**H, H'**) MIP of *CLE40* in a *wus-7* background shows expression through the entire IFM and in the centre of flower primordia. The longitudinal optical section (**H'**) reveals that *CLE40* is also expressed in the CZ as well as in the OC of the IFM (N = 12). Dashed orange lines indicate the planes of longitudinal sections, dashed red line in (**A**), (**B**), (**E**) and (**F**) marks the IFM area, the dashed pink line in (**A**) and (**B**) marks the sepals, the dashed rose line in (**A**) and (**B**) marks the FMs, the dashed white line in (**E'**) marks the CZ. Scale bars: 50 µm (**A, B, G, H**), 20 µm (**A', B', E, E', F, F', G', H'**), 1 mm (**C, D**), 500 µm (**D'**). PI: propidium iodide; L1: layer 1 projection; C: cotyledon; LP: leaf primordium.

The online version of this article includes the following figure supplement(s) for figure 3:

**Figure supplement 1.** *CLE40* expression is lacking in the central zone (CZ) and organizing centre (OC).

**Figure supplement 2.** *CLE40* expression is extended in *wus-7* mutants.

between CLV1 and BAM1 is not symmetrical since CLV1 is expressed in a wild-typic pattern in *bam1-3* mutants (***Figure 8—figure supplement 4***). Meristem sizes of *bam1-3* mutants reached 70% of the wild type, and double mutants of *cle40-2;bam1-3* did not differ significantly. However, double mutants of *cle40-2;clv1-101* developed like the *clv1-101* single mutant, indicating an epistatic relationship. Importantly, both *clv1-101* and *bam1-3* mutants lack BAM1 function in the meristem periphery (***Figure 5F***), where also *CLE40* is highly expressed, which could explain the observed epistatic relationships of *cle40-2* with both *clv1-101* and *bam1-3*. Similar genetic relationships for *CLV3, CLE40, CLV1* and *BAM1* were noticed when analysing carpel number as a proxy for FM sizes. We also noted that generation of larger IFMs and FMs in different mutants was negatively correlated with leaf size, which we cannot explain so far (***Figure 1—figure supplement 1***). In the root meristem, we found that the BAM1 receptor, but not CLV1, is required for CLE40-dependent root meristem development, suggesting that CLE40 can act through BAM1 (***Figure 6—figure supplement 1***).

For the shoot, we hypothesize that CLE40 signals from the meristem periphery via BAM1 to promote meristem growth. Next, we aimed to determine if the commonalities between *cle40-2* and *bam1-3* mutants extend beyond their effects on meristem size.

## A CLE40 and BAM1 signalling pathway promotes *WUS* expression in the meristem periphery

We next analysed the number of *WUS*-expressing cells in wild type and mutant meristems using a *WUS:NLS-GFP* transcriptional reporter. Compared to wild type, the *WUS* expression domain was laterally strongly expanded in both *clv3-9* and *clv1-101*. Interestingly, *WUS* signal extended also into the L1 layer of *clv1-101*, albeit in a patchy pattern (***Figure 7A–C' and F***, ***Figure 7—figure supplement 1***, ***Figure 7—figure supplement 2***). Also noteworthy is that *BAM1* was expressed at a higher level in the L1 layer of *clv1* mutants. *cle40-2* mutants showed a reduction in the number of *WUS*-expressing cells down to ~50% wild-type levels (***Figure 7D–D' and F***, ***Figure 7—figure supplement 1***, ***Figure 7—figure supplement 2***). Importantly, *WUS* remained expressed in the centre of the meristem, but was found in a narrow domain. In *bam1-3* mutants, the *WUS* domain was similarly reduced as in *cle40-2*, and *WUS* expression focussed in the meristem centre (***Figure 7E, E' and F***, ***Figure 7—figure supplement 1***, ***Figure 7—figure supplement 2***). In contrast, both *clv3-9* and *clv1-101* mutants express *WUS* in a laterally expanded domain (***Figure 7B' and C'***, ***Figure 7—figure supplement 1***, ***Figure 7—figure supplement 2***).

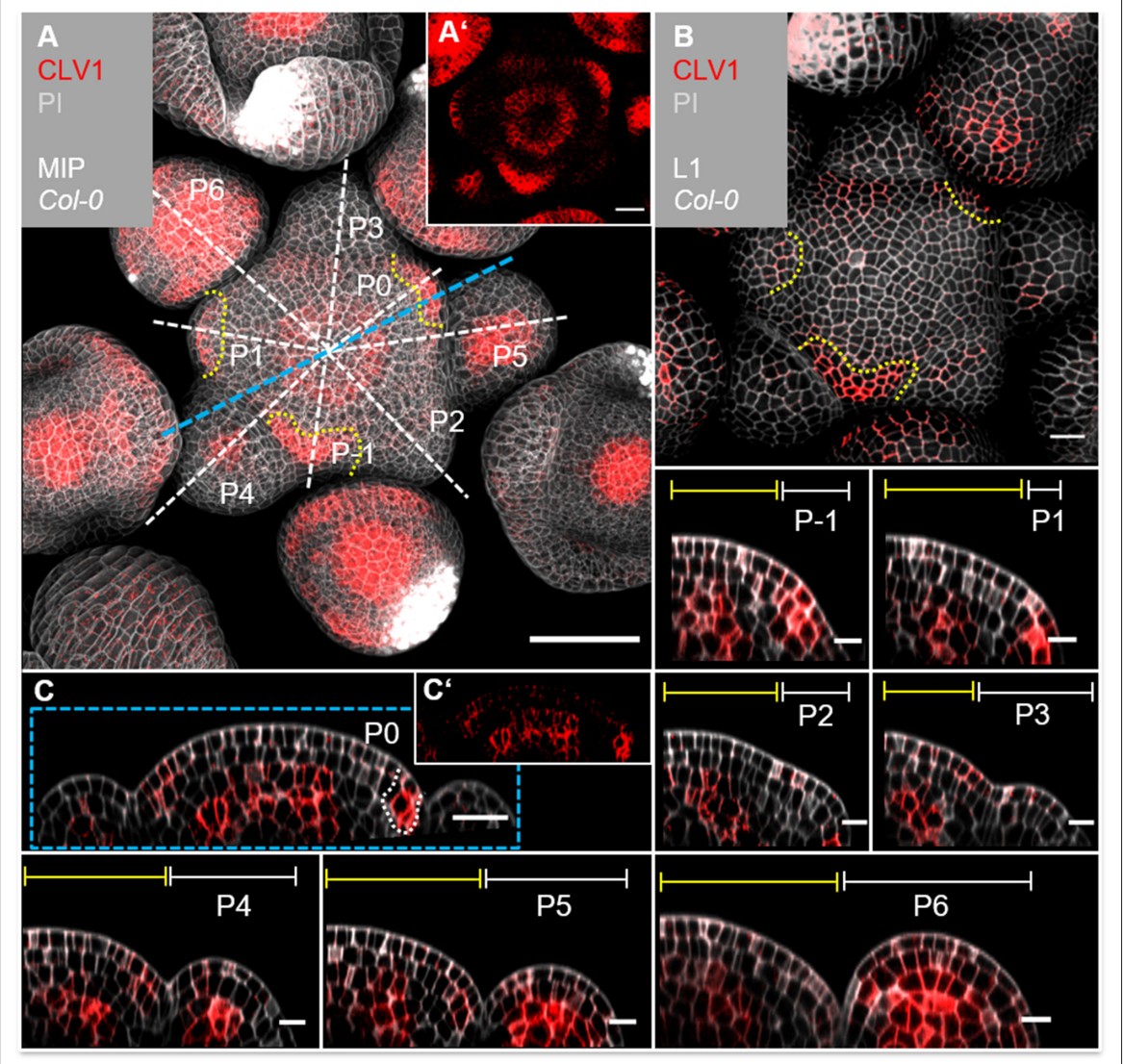

**Figure 4.** *CLV1* is expressed in the organizing centre (OC) and cells of incipient organ primordia. (**A**) Maximum intensity projection (MIP) of *CLV1* under its endogenous promoter (*CLV1:CLV1-GFP//Col-0*) at 5 weeks after germination (WAG) shows *CLV1* expression in the OC of the meristems, inflorescence meristem (IFM) and flower meristems (FMs), in incipient organ primordia (P-1–P1) and in sepals (N = 15). (**A'**) MIP of the IFM from (**A**) without propidium iodide (PI) staining. (**B**) In the layer 1 (L1) projection, *CLV1* expression is detected in cells of incipient organs. (**C**) Longitudinal section through the IFM shows *CLV1* expression in the OC and P0. (**C'**) XZ section from (**C**) without PI staining. (P-1–P6) *CLV1* expression is detected in incipient organ primordia in L1 and L2 (-P1, P0), in the L2 of P1 and in the OC of the IFM and FMs from P4 to P6. Dashed white and blue lines indicate the planes of longitudinal sections, yellow dashed lines in (**A**) and (**B**) mark incipient organ primordia (P-1–P1), yellow lines (P-1–P6) indicate the IFM region, white lines mark the primordium. Scale bars: 50 μm (**A**), 20 μm (**B, C**), 10 μm (P1–P6), P: primordium.

The online version of this article includes the following figure supplement(s) for figure 4:

**Figure supplement 1.** The translational *CLV1:CLV1-GFP* reporter line rescues the carpel and shoot phenotype of *clv1-101* mutants.

**Figure supplement 2.** Two independent translational *CLV1* reporter lines show exactly the same expression pattern.

**Figure supplement 3.** *CLV1* expression in multiple inflorescence meristem (IFMs).

To integrate our finding that *CLE40* expression is repressed by WUS activity with the observation that *WUS*, in turn, is promoted by *CLE40* signalling, we hypothesize that the *CLE40-BAM1-WUS* interaction establishes a new negative feedback loop. The *CLE40-BAM1-WUS* negative feedback loop acts in the meristem periphery, while the *CLV3-CLV1-WUS* negative feedback loop acts in the meristem centre along the apical-basal axis. Both pathways act in parallel during development to regulate the

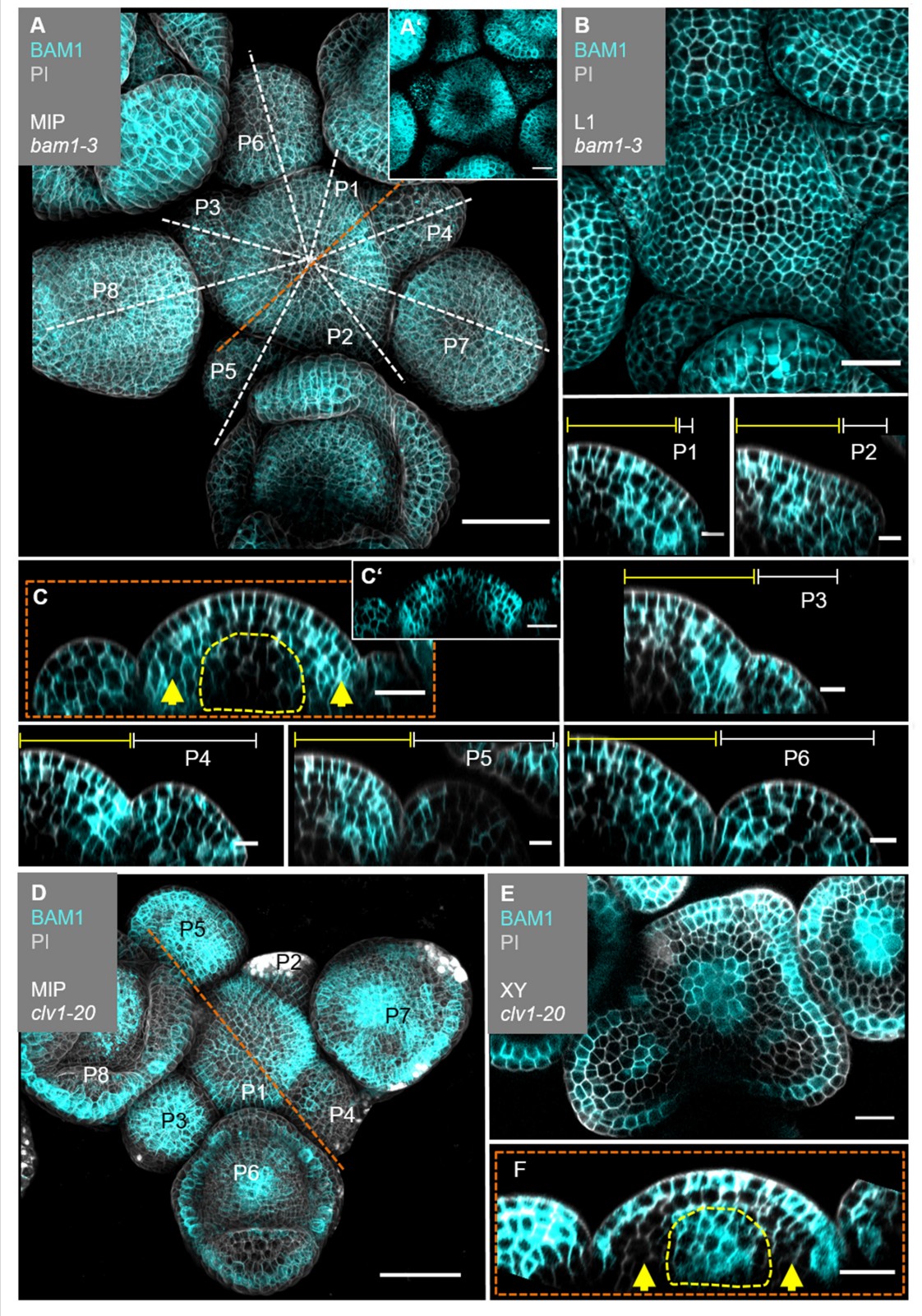

**Figure 5.** *BAM1* expression is elevated in the flanks of the inflorescence meristem (IFM) and not detectable in the organizing centre (OC). (**A**) Maximum intensity projection (MIP) of *BAM1* under its endogenous promoter (*BAM1:BAM1-GFP//bam1-3*) at 5 weeks after germination (WAG). *BAM1* expression is detected nearly throughout the entire inflorescence (IFM, flower meristem [FM], sepals) with weak expression in the central zone (CZ) of IFM and FMs (N = 15). (**A'**) MIP of the IFM from (**A**) without propidium iodide (PI) staining. (**B**) The layer 1 (L1) projection of the IFM shows ubiquitous expression of

*Figure 5 continued on next page*

*Figure 5 continued*

*BAM1*. (**C**) Longitudinal optical section through the IFM shows elevated *BAM1* expression in the flanks (yellow arrows) and a lack of *BAM1* expression in the OC. (**C'**) XZ section from (**C**) without PI staining. (P1–P6) *BAM1* expression is found in all primordia cells. (**D**) MIP of *BAM1* in a *clv1-20* mutant (*BAM1:BAM1-GFP//bam1-3;clv1-20*). *BAM1* expression is detected in most parts of the inflorescence, especially in the centre of the IFM and FMs (N = 9). (**E**) Cross-section (XY) of the IFM (from **D**) shows *BAM1* expression in a *clv1-20* mutant in the CZ (IFM and FMs) and the L1/L2. (**F**) Longitudinal optical section through the meristem (from **D**) shows *BAM1* expression in the OC and the L1, while no *BAM1* expression is detected in the peripheral zone (PZ) (yellow arrows). Dashed white and orange lines indicate longitudinal sections; dashed yellow lines in (**C**) and (**F**) mark the OC area, yellow lines (P1–P6) indicate the IFM region, white lines (P1–P6) mark the primordium and yellow arrows indicate high (**C**) or no (**F**) *BAM1* expression in the PZ. Scale bars: 50 μm (**A, D**), 20 μm (**B, C, E, F**), 10 μm (P1–P6). P: primordium.

The online version of this article includes the following figure supplement(s) for figure 5:

**Figure supplement 1.** The translational *BAM1:BAM1-GFP* reporter line rescues the shoot phenotype of the double mutant *bam1-3;clv1-20*.

**Figure supplement 2.** The translational BAM1 reporter line shows similar expression in three independent T1 lines.

**Figure supplement 3.** Expression of the translational BAM1 reporter (*BAM1:BAM1-GFP*) shifts from the PZ in *bam1-3* mutants to the organizing centre (OC) and L1 in *bam1-3;clv1-20* double mutants.

**Figure supplement 4.** Quantification of expression pattern of the translational BAM1 reporter line in *bam1-3* (N = 9) and *bam1-3;clv1-20* (N = 9) mutants.

**Figure supplement 5.** BAM1 and CLV1 are receptors for CLE40 and CLV3, respectively.

**Figure supplement 6.** Expression patterns of *CLE40* and *BAM1* overlap in the inflorescence meristem (IFM).

size of the *WUS* expression domain in the meristem, possibly by perceiving input signals from two different regions, the CZ and the PZ, of the meristem.

We then asked how the two signalling pathways converge on the regulation of *WUS* expression, control meristem growth and development. So far, we showed that both *CLV3-CLV1* and *CLE40-BAM1* signalling control meristem size, but in an antagonistic manner. However, we noticed that the different mutations in peptides and receptors affected distinct aspects of meristem shape. We therefore analysed meristem shape by measuring meristem height (the apical-basal axis) at its centre, and meristem diameter (the radial axis) at the base in longitudinal sections. The ratio of height to width then gives a shape parameter 'σ' (from the Greek word σχῆμα = shape). In young IFMs at 4–5 WAG, when inflorescence stems were approximately 5–8 cm long, meristems of *cle40-2* and *bam1-3* mutants were slightly reduced in width, and strongly reduced in height, resulting in reduced σ in comparison to *Col-0* (**Figure 7G**, **Figure 7—figure supplement 3**). Meristems of *clv1-101* and *clv3-9* mutants were similar in width to wild type, but strongly increased in height, giving high σ values (**Figure 7**, **Figure 7—figure supplement 3**). This indicates that *CLV3-CLV1* signalling mostly restricts meristem growth along the apical-basal axis, while *CLE40-BAM1* signalling promotes meristem growth along both axes.

Our data expand the current model of shoot meristem homeostasis by taking into account that stem cells are lost from the OC during organ initiation in the PZ (**Figure 8**). CLV3 signals from the CZ via CLV1 in the meristem centre to confine *WUS* expression to the OC. The diffusion of WUS protein along the apical-basal axis towards the meristem tip establishes the CZ and activates *CLV3* expression as a feedback signal. During plant growth, rapid cell division activity and organ initiation requires the replenishment of PZ cells from the CZ, which can be mediated by increased *WUS* activity. We now propose that the PZ generates CLE40 as a short-range or autocrine signal that acts through BAM1 in the meristem periphery. Since *BAM1* and *WUS* expression does not overlap, we postulate the generation of a diffusible factor that relies on *CLE40-BAM1* and acts from the PZ to promote *WUS* expression. WUS, in turn, represses *CLE40* expression from the OC, thus establishing a second negative feedback regulation. Together, the two intertwined pathways serve to adjust WUS activity in the OC and incorporate information on the actual size of the stem cell domain, via *CLV3-CLV1*, and the growth requirements from the PZ via *CLE40-BAM1*.

## Discussion

Shoot meristems are the centres of growth and organ production throughout the life of a plant. Meristems fulfil two main tasks, which are the maintenance of a non-differentiating stem cell pool, and the assignment of stem cell daughters to lateral organ primordia and differentiation pathways (*Hall and Watt, 1989*). Shoot meristem homeostasis requires extensive communication between the

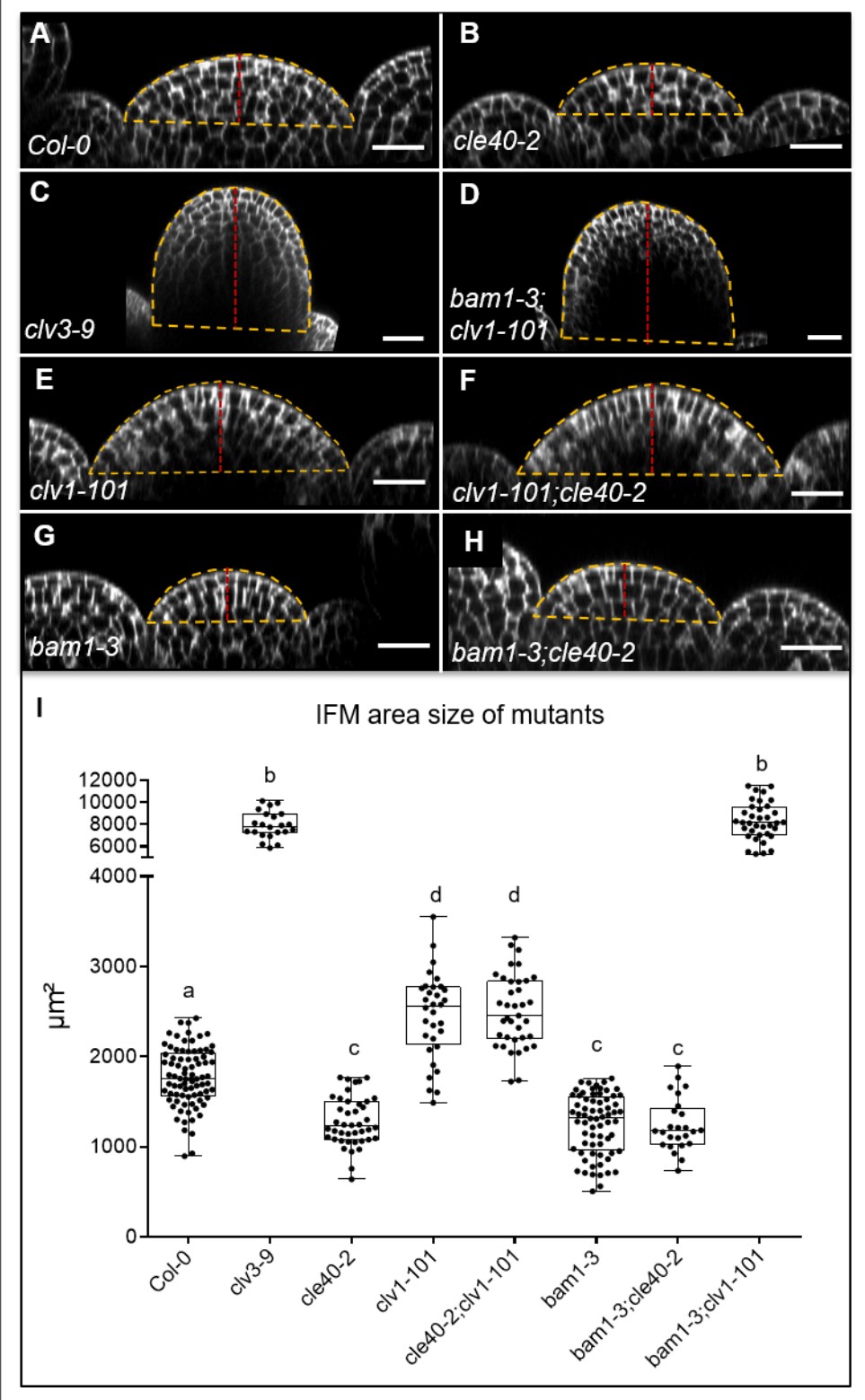

**Figure 6.** Inflorescence meristem (IFM) size of single and double mutants. XZ sections through the centre of IFMs of (**A**) *Col-0*, (**B**) *cle40-2*, (**C**) *clv3-9*, (**D**) *bam1-3;clv1-101*, (**E**) *clv1-101*, (**F**) *clv1-101;cle40-2*, (**G**) *bam1-3* and (**H**) *bam1-3;cle40-2* plants. (I) Box and whisker plot of the IFM area size of *Col-0* (N = 82), various single (*clv3-9* [N = 22], *cle40-2* [N = 42], *clv1-101* [N = 32], *bam1-3* [N = 68]) and double mutants (*cle40-2;clv1-101* [N = 37], *cle40-2;bam1-3* [N = 25] and *bam1-3;clv1-101* [N = 36]) at 6 weeks after germination (WAG).The yellow dashed line depicts the

*Figure 6 continued on next page*

*Figure 6 continued*

area of the meristem that was measured, and the dashed red line indicates the height of the meristems. Scale bar: 20 µm (**A–H**). Statistical groups were assigned after calculating p-values by ANOVA and Tukey's multiple comparison test (differential grouping from p≤0.01).

The online version of this article includes the following figure supplement(s) for figure 6:

**Figure supplement 1.** *bam1-3* and *bam1-3;clv1-101* mutants are resistant to CLE40 peptide treatment.

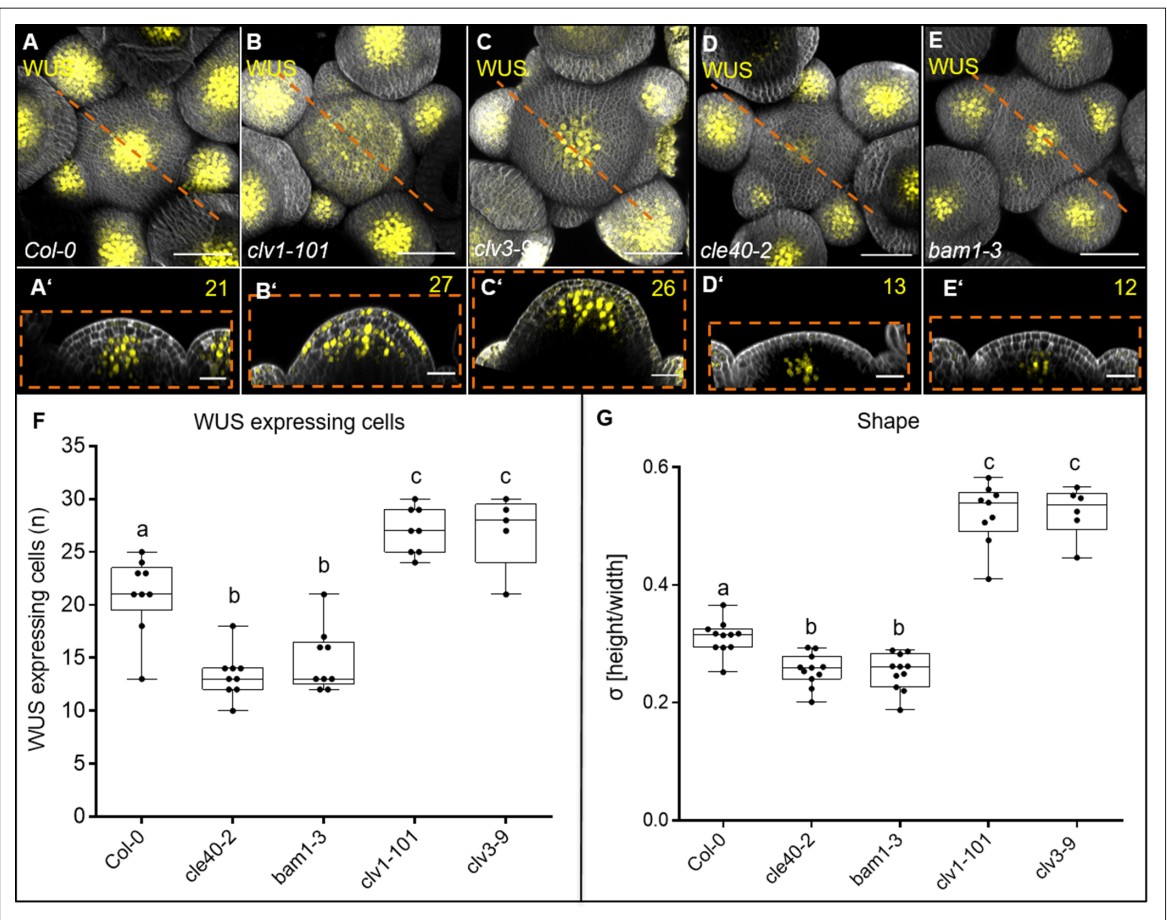

**Figure 7.** CLE40 and BAM1 promote *WUS* expression. (**A–E'**) Maximum intensity projection (MIP) and longitudinal optical section of inflorescences at 5 weeks after germination (WAG) expressing the transcriptional reporter *WUS:NLS-GFP* in a (**A, A'**) *Col-0*, (**B, B'**) *clv1-101*, (**C, C'**) *clv3-9*, (**D, D'**) *cle40-2* and (**E, E'**) *bam1-3* background. In (**A**) wild-type plants, the WUS domain is smaller compared to the expanded WUS domain in (**B**) *clv1-101* and (**C**) *clv3-9* mutants. The WUS domain of (**D**) *cle40-2* and (**E**) *bam1-3* mutants is decreased compared to wild-type plants. Longitudinal optical sections of (**B'**) *clv1-101* and (**C'**) *clv3-9* mutants expand along the basal-apical axis while the meristem shape of (**D'**) *cle40-2* and (**E'**) *bam1-3* mutants is flatter compared to (**A'**) wild-type plants,. (**F**) Box and whisker plot shows the number of *WUS*-expressing cells in a single plane through the organizing centre (OC) of inflorescence meristems (IFMs) of *Col-0* (N = 9), *cle40-2* (N = 9), *bam1-3* (N = 9), *clv1-101* (N = 8) and *clv3-9* (N = 5). (**G**) At 5 WAG, *bam1-3* (N = 11) and *cle40-2* (N = 11) mutants have flatter meristems than wild-type plants (decreased σ value compared to *Col-0* [N = 11]), while *clv1-101* [N = 9] and *clv3-9* [N = 6] mutants increase in their IFM height showing a higher σ value. Scale bars: 50 µm (**A–E**), 20 µm (**A'–E'**), Statistical groups and stars were assigned after calculating p-values by ANOVA and Tukey's multiple comparison test (differential grouping from p≤0.01). yellow numbers: *WUS*-expressing cells in the CZ; σ value: height/width of IFMs.

The online version of this article includes the following figure supplement(s) for figure 7:

**Figure supplement 1.** Number of *WUS*-expressing cells in the inflorescence meristem (IFM) of various mutant backgrounds detected with Imaris software.

**Figure supplement 2.** Number of *WUS*-expressing cells in a longitudinal section through the meristem in multiple inflorescence meristems (IFMs).

**Figure supplement 3.** Inflorescence meristem (IFM) height, width and shape at 5 weeks after germination (WAG).

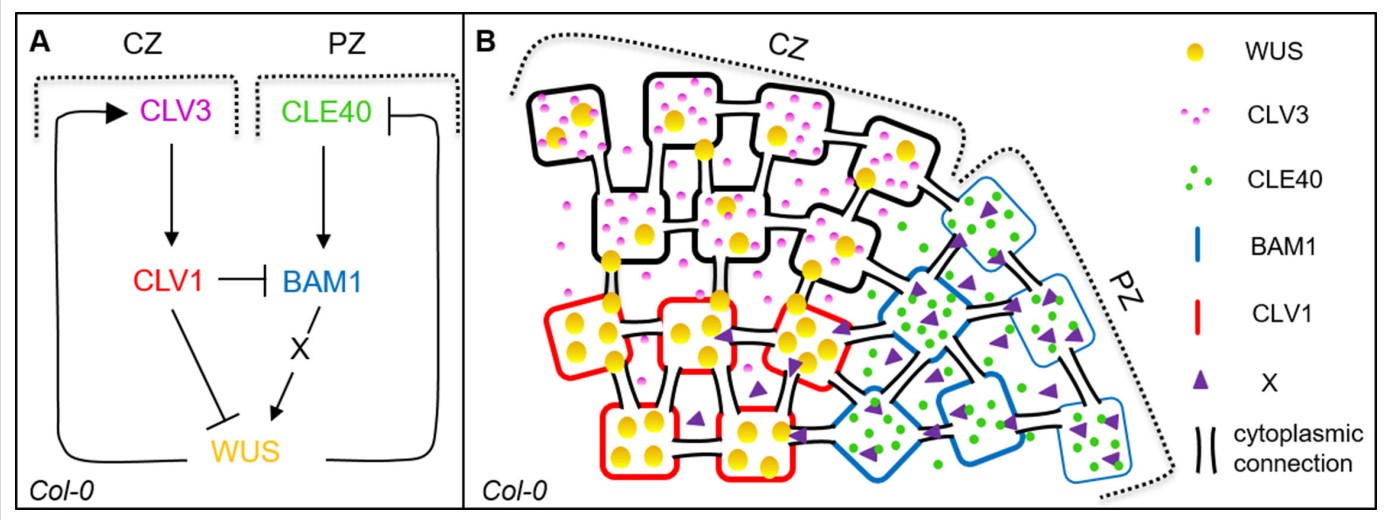

**Figure 8.** Schematic model of two intertwined signalling pathways in the shoot meristem. (**A, B**) Schematic representation of two negative feedback loops in the inflorescence meristem (IFM) of *Arabidopsis thaliana*. CLV3 in the central zone (CZ) binds to the LRR receptor CLV1 to activate a downstream signalling cascade which leads to the repression of the transcription factor WUS. In a negative feedback loop WUS protein moves to the stem cells to activate *CLV3* gene expression. In the peripheral zone (PZ) of the IFM, a second negative feedback loop controls meristem growth by CLE40 and its receptor BAM1. CLE40 binds to BAM1 in an autocrine manner, leading to the activation of a downstream signal 'X' which promotes WUS activity. WUS protein in turn represses the expression of the *CLE40* gene. Arrows indicate a promoting effect, and the blocked line indicates a repressing signal.

The online version of this article includes the following figure supplement(s) for figure 8:

**Figure supplement 1.** Schematic model of the two intertwined signalling pathways in a *clv1-101* mutant background.

**Figure supplement 2.** Schematic model of the intertwined signalling pathways in a *clv3-9* mutant background.

**Figure supplement 3.** Schematic model of the intertwined signalling pathways in a *cle40-2* mutant background.

**Figure supplement 4.** Schematic model of the intertwined signalling pathways in a *bam1-3* mutant background.

CZ, the OC and the PZ. The discovery of CLV3 as a signalling peptide, which is secreted exclusively from stem cells in the CZ, and its interaction with WUS in a negative feedback loop was fundamental to our understanding of such communication pathways (*Fletcher, 2020*). Here, we analysed the function of *CLE40* in shoot development of *Arabidopsis* and found that *WUS* expression in the OC is under positive control from the PZ due to the activity of a CLE40-BAM1 signalling pathway. IFM size is reduced in *cle40* mutants, indicating that *CLE40* signalling promotes meristem size. Importantly, *CLE40* is expressed in the PZ, late-stage FMs and differentiating organs. A common denominator for the complex and dynamic expression pattern is that *CLE40* expression is confined to meristematic tissues, but not in organ founder sites or in regions with high WUS activity, such as the OC and the CZ. Both misexpression of *WUS* in the *CLV3* domain (*Figure 3F*), studies of *clv3* mutants with expanded stem cell domains (*Figure 3B*, *Figure 3—figure supplement 1*) and analysis of *wus* mutants (*Figure 3*, *Figure 3—figure supplement 2*) underpinned the notion that *CLE40* expression, in contrast to *CLV3*, is negatively controlled in a *WUS*-dependent manner. Furthermore, we found that the number of *WUS*-expressing cells in *cle40* mutant IFMs is strongly reduced, indicating that *CLE40* exerts its positive effects on IFM size by expanding the *WUS* expression domain.

So far, the antagonistic effects of *Arabidopsis CLV3* and *CLE40* on meristem size can only be compared to the antagonistic functions of *MpCLE1* and *MpCLE2* on the gametophytic meristems of *M. polymorpha*, which signal through two distinct receptors, MpTDR and MpCLV1, respectively (*Hata and Kyozuka, 2021*). By the complementation of *clv3* mutants through expression of *CLE40* from the *CLV3* promoter, it was shown previously that *CLE40* and *CLV3* are able to activate the same downstream receptors (*Hobe et al., 2003*). Our detailed analysis of candidate receptor expression patterns showed that *CLV3* and *CLV1* are expressed in partially overlapping domains in the meristem centre, while *CLE40* and *BAM1* are confined to the meristem periphery. Like *cle40* mutants, *bam1* mutant IFMs are smaller and maintain a smaller *WUS* expression domain, supporting the notion that *CLE40* and *BAM1* comprise a signalling unit that increases meristem size by promoting *WUS* expression. The

antagonistic functions of the *CLV3-CLV1* and *CLE40-BAM1* pathways in the regulation of *WUS* are reflected in their complementary expression patterns. There is cross-regulation between these two signalling pathways at two levels: (1) *WUS* has been previously shown to promote *CLV3* levels in the CZ, and we here show that *WUS* represses (directly or indirectly) *CLE40* expression in the OC and in the CZ (**Figure 3B**, **Figure 3—figure supplement 1**) and (2) *CLV1* represses *BAM1* expression in the OC, and thereby restricts *BAM1* to the meristem periphery (**Figures 5 and 6**). In *clv1* mutants, *BAM1* shifts from the meristem periphery to the OC, and the *WUS* domain laterally expands in the meristem centre (**Figures 5F and 7B'**). Furthermore, *BAM1* expression increases also in the L1, which could cause the observed irregular expression of *WUS* in the outermost cell layer of *clv1* mutants. The role of BAM1 in the OC is not entirely clear: despite the high-sequence similarity between CLV1 and BAM1, the expression of *BAM1* in the OC is not sufficient to compensate for the loss of CLV1 (**Figure 5D–F**, **Nimchuk et al., 2015**). In the OC, BAM1 appears to restrict *WUS* expression to some extent since *clv1;bam1* double mutants reveal a drastically expanded IFM (**Deyoung and Clark, 2008**). However, it is possible that BAM1 in the absence of CLV1 executes a dual function: to repress *WUS* in response to CLV3 in the OC as a substitute for CLV1 and simultaneously to promote *WUS* expression in the L1 in response to CLE40.

The expression domains of *CLE40* and its receptor *BAM1* largely coincide, suggesting that CLE40 acts as an autocrine signal. Similarly, protophloem sieve element differentiation in roots is inhibited by CLE45, which acts as an autocrine signal via BAM3 (**Kang and Hardtke, 2016**). Since *WUS* is not expressed in the same cells as *BAM1*, we have to postulate a non-cell-autonomous signal X that is generated in the PZ due to *CLE40-BAM1* signalling and diffuses towards the meristem centre to promote *WUS* expression (**Hohm et al., 2010**). As a result, *CLE40-BAM1* signalling from the PZ will provide the necessary feedback signal that stimulates stem cell activity and thereby serves to replenish cells in the meristem for the initiation of new organs. The *CLV3-CLV1* signalling pathway then adopts the role of a necessary feedback signal that avoids an excessive stem cell production.

The two intertwined, antagonistically acting signalling pathways that we described here allow us to better understand the regulation of shoot meristem growth, development and shape. The previous model, which focussed mainly on the interaction of the CZ and the OC via the *CLV3-CLV1-WUS* negative feedback regulation, lacked any direct regulatory contribution from the PZ. *EPFL* peptides were shown to be expressed in the periphery and to restrict both *CLV3* and *WUS* expression via ER (**Zhang et al., 2021**). However, *EPFL* peptide expression is not reported to be feedback regulated from the OC or CZ, and the main function of the *EPFL-ER* pathway is therefore to restrict overall meristem size (**Zhang et al., 2021**). The second negative feedback loop controlled by CLE40, which we uncovered here, enables the meristem to fine-tune stem cell activities in response to fluctuating requirements for new cells during organ initiation. Due to the combined activities of CLV3 and CLE40, the OC (with WUS as a key player) can now record and compute information from both, the CZ and PZ. Weaker *CLV3* signalling, indicating a reduction in the size of the CZ, induces preferential growth of the meristem along the apical-basal axis (increasing σ), while weaker CLE40 signals, reporting a smaller PZ,

**Table 1.** Mutants analysed in this study.

| Allele | Gene | Mutation | Reference | Background |
|---|---|---|---|---|
| bam1-3 | AT5G65700 | T-DNA | *Alonso et al., 2003*; SALK_015302 | Col-0 |
| cle40-2 | AT5G12990 | Transposon mutation | *Stahl et al., 2009* | Col-0 |
| cle40-cr1 | | | | |
| cle40-cr2 | | | | |
| cle40-cr3 | AT5G12990 | CRISPR | *Yamaguchi et al., 2017* | Col-0 |
| clv3-9 | AT2G27250 | EMS | *Brand et al., 2000* | Col-0 |
| clv1-20 | AT1G75820 | T-DNA | SALK_008670 | Col-0 |
| clv1-101 | AT1G75820 | T-DNA | *Kinoshita et al., 2010*; CS858348 | Col-0 |
| wus-7 | AT2G17950 | EMS | *Graf et al., 2010* | L.er. |

**Table 2.** Primers and methods used for genotyping.

| Allele | Method | Primer | PCR product |
|---|---|---|---|
| bam1-3 | PCR | bam1-3_F: ctaacgactctccgggagct<br>bam1-3_R: taaggaccacagagatcaggattac<br>LbaI_R: tggttcacgtagtgggccatcg | WT amp.: 1208 bp<br>mutant amp.: 998 bp |
| cle40-2 | dCAPS | cle40-2_F: GGAGAAACACAAGATACGAAAGCCATG<br>cle40-2_R: ATTGTGATTTGATACCAACTTAAAA | Restriction enzyme: AseI<br>WT amp.: 460 + 200 bp<br>mutant amp.: 410 + 200 + 60 bp |
| cle40-cr1 | | | |
| cle40-cr2 | | | Restriction enzyme: BamHI<br>WT amp.: 750 bp |
| cle40-cr3 | dCAPS | cle40-cr_F: ATGGCGGCGATGAAATACAA<br>cle40-cr_R: GTTACGCTTTGGCATCTTTCC | mutant amp.: 491 + 259 bp |
| clv1-20 | PCR | clv1-20_F: TTTGAATAGTGTGTGACCAAATTTGA<br>clv1-20_R: TCCAATGGTAATTCACCGGTG<br>LBa.1: TGGTTCACGTAGTGGGCCATCG | WT amp.: 860 bp<br>mutant amp: 1200 bp |
| clv1-101 | PCR | clv1-101_F: TTCTCCAAATTCACCAACAGG<br>clv1-101_R: CAACGGAGAAATCCCTAAAGG<br>WiscLox_LT6_R: AATAGCCTTTACTTGAGTTGGCGTAAAAG | WT amp.: 1158 bp<br>mutant amp.: 896 bp |
| wus-7 | dCAPS | wus-7_F: CCGACCAAGAAAGCGGCAACA<br>wus-7_R: AGACGTTCTTGCCCTGAATCTTT | Restriction enzyme: XmnI<br>WT amplification: 216 bp<br>mutant amp.: 193 + 23 bp |

would decrease σ and flatten meristem shape. It will be intriguing to investigate if different levels of CLV3 and CLE40 also contribute to the shape changes that are observed during early vegetative development or upon floral transition in *Arabidopsis*.

Many shoot-expressed CLE peptides are encoded in the genomes of maize, rice and barley, which could act analogously to CLV3 and CLE40 of *Arabidopsis*. It is tempting to speculate that in grasses a CLE40-like, stem cell-promoting signalling pathway is more active than a CLV3-like, stem cell-restricting pathway. This could contribute to the typical shape of cereal SAMs, which are, compared to the dome-shaped SAM of dicotyledonous plants, extended along the apical-basal axis.

## Materials and methods
### Plant material and growth conditions

All wild-type *Arabidopsis thaliana* (L.) Heynh. plants used in this study are ecotype Columbia-0 (*Col-0*), except for *wus-7* mutants which are in Landsberg *erecta* (L.*er*.) background. Details about *A. thaliana* plants carrying mutations in the following alleles – *bam1-3, cle40-2, cle40-cr1, cle40-cr2, cle40-cr3, clv1-101, clv3-9* and *wus-7* – are described in *Table 1*. All mutants are in *Col-0* background and are assumed to be null mutants, except for *wus-7* mutants. *cle40* mutants (*cle40-2, cle40-cr1, cle40-cr2, cle40-cr3*) have either a stop codon, a T-DNA insertion or deletion in or before the crucial CLE box domain. *clv3-9* mutants were generated in 2003 by the lab of R. Simon. *clv3-9* mutants were created by EMS, resulting in a W62STOP mutation before the critical CLE domain region. *bam1-3* and *clv1-101* mutants have been described as null mutants before (*DeYoung et al., 2006*; *Kinoshita et al., 2010*), while *clv1-20* is a weak allele which contains a insertion within the 5'-UTR of CLV1 and results in a reduced mRNA level (*Durbak and Tax, 2011*). *wus-7* is a weak allele and mutants were described in previous publications (*Graf et al., 2010*). Double mutants were obtained by crossing the single mutant plants until both mutations were proven to be homozygous for both alleles. Genotyping of the plants was performed either by PCR or dCAPS method with the primers and restrictions enzymes listed in *Table 2*.

Before sowing, seeds were either sterilized for 10 min in an ethanol solution (80% v/v ethanol, 1.3% w/v sodium hypochloride, 0.02% w/v SDS) or for 1 hr in a desiccator in a chloric gas atmosphere (50 mL of 13% w/v sodium hypochlorite with 1 mL 37% HCl). Afterwards, seeds were stratified for 48 hr at 4°C in darkness. Seeds on soil were then cultivated in phytochambers under long day (LD) conditions (16 hr light/8 hr dark) at 21°C. For selection of seeds or imaging of vegetative meristems, seeds were sown on ½ Murashige & Skoog (MS) media (1% w/v sucrose, 0.22% w/v MS salts + B5

vitamins, 0.05% w/v MES, 12 g/L plant agar, adjusted to pH 5.7 with KOH) in squared Petri dishes. Seeds in Petri dishes were kept in phytocabinets under continuous light conditions at 21°C and 60% humidity.

## Cloning of reporter lines

The CLE40 (*CLE40:Venus-H2B*) reporter line was cloned using the Gateway method (*Curtis and Grossniklaus, 2003*). The vector *CLE40:Venus-H2B* carries a 2291 bp long DNA fragment extending 5′ from the translational start codon of *CLE40* that drives the expression of a *Venus-H2B* fusion protein. The DNA fragment was amplified via PCR using the oligonucleotides proCLE40_F and proCLE40_R (Table 5). As PCR template, wild-type *Col-0* DNA was used. The fragment was inserted in

**Table 3.** Entry vectors used for cloning.

| Name | Description | Bacterial resistance | Backbone | Reference/origin |
|---|---|---|---|---|
| proBAM1 (pGD288) | BAM1 promoter 3522 bp upstream from transcription start | Ampicillin | pGGA000 | Grégoire Denay |
| proCLE40 | CLE40 promoter 2291 bp upstream from translational start codon | Kanamycin | pENTR/D-TOPO | Rene Wink |
| proCLV3 | CLV3 promoter 1480 bp upstream from transcription start | Ampicillin | pGGA000 | Jenia Schlegel |
| proCLV1 | CLV1 promoter 5759 bp upstream from transcription start | Ampicillin | pGGA000 | Patrick Blümke |
| omega-element (pGGB002) | Omega-element | Ampicillin | pGGB000 | *Lampropoulos et al., 2013* |
| SV40 NLS (pGGB005) | SV40 NLS (SIMIAN VIRUS 40 NUCLEAR LOCALIZATION SIGNAL) | Ampicillin | pGGB000 | *Lampropoulos et al., 2013* |
| BAM1_CDS (pGD351) | BAM1 coding region genomic region of BAM1 START to one codon before STOP, including introns, internal BsaI sites removed | Ampicillin | pGGC000 | Grégoire Denay |
| CLV1_CDS | CLV1 coding region 2946 bp coding region amplified from genomic Col-0 DNA without STOP codon and internal BsaI site removed | Ampicillin | pGGC000 | Jenia Schlegel |
| 3x-mCherry (pGGC026) | 3x mCherry | Ampicillin | pGGC000 | *Lampropoulos et al., 2013* |
| linker-GFP (pGD165) | linker(10aa)-eGFP | Ampicillin | pGGD000 | Grégoire Denay |
| d-dummy (pGGD002) | d-dummy | Ampicillin | pGGD000 | *Lampropoulos et al., 2013* |
| tCLV3 | CLV3 terminator 1257 bp downstream of transcription stop | Ampicillin | pGGE000 | Jenia Schlegel |
| tUBQ10 (pGGE009) | UBQ10 terminator | Ampicillin | pGGE000 | *Lampropoulos et al., 2013* |
| BastaR (pGGF008) | pNOS:BastaR (chi sequence removed):tNOS | Ampicillin | pGGF000 | *Lampropoulos et al., 2013* |
| D-AlaR (pGGF003) | pMAS:D-AlaR:tMAS | Ampicillin | pGGF000 | *Lampropoulos et al., 2013* |

**Table 4.** Destination vectors used to generate transgenic *A. thaliana* reporter lines.

| Name | Backbone | Promoter | N-tag | CDS | C-tag | Terminator | Resistance |
|------|----------|----------|-------|-----|-------|------------|------------|
| | | | | | | tUBQ10 | |
| *BAM1:BAM1-GFP* | pGGZ001 | proBAM1 | Ω-element (pGGB002) | BAM1-CDS | linker-GFP (pGD165) | (pGGE009) | D-Alanin (pGGF003) |
| *CLE40:Venus-H2B* | pMDC99 | proCLE40 | - | Venus | H2B | T3A | Hygromycin |
| | | | | | | tUBQ10 | |
| *CLV1:CLV1-GFP* | pGGZ001 | proCLV1 | Ω-element (pGGB002) | CLV1-CDS | linker-GFP (pGD165) | (pGGE009) | BastaR (pGGF008) |
| *CLV3:NLS-3xmCherry* | pGGZ001 | proCLV3 | SV40 NLS | 3x-mCherry (pGGC026) | d-dummy (pGGD002) | tCLV3 | BastaR (pGGF008) |

pENTR-D-TOPO via directional TOPO-cloning. The insert was then transferred into a modified plant transformation vector pMDC99 containing the *Venus-H2B* sequence (*Curtis and Grossniklaus, 2003*).

CLV1 (*CLV1:CLV1-GFP*), BAM1 (*BAM1:BAM1-GFP*) and CLV3 (*CLV3:NLS-3xmCherry*) reporter lines were cloned using the GreenGate method (*Lampropoulos et al., 2013*). Entry and destination plasmids are listed in *Table 3* and *Table 4*. Promoter and coding sequences were PCR amplified from genomic *Col-0* DNA which was extracted from rosette leaves of *Col-0* plants. Primers used for amplification of promoters and coding sequences can be found in *Table 5* with the specific overhangs used for the GreenGate cloning system. Coding sequences were amplified without the stop codon to allow transcription of fluorophores at the C-terminus. BsaI restriction sites were removed by site-directed mutagenesis using the 'QuikChange II Kit' following the manufacturer's instructions (Agilent Technologies). Plasmid DNA amplification was performed by heat-shock transformation into *Escherichia coli* DH5α cells (10 min on ice, 1 min at 42°C, 1 min on ice, 1 hr shaking at 37°C), which were subsequently plated on selective LB medium (1% w/v tryptone, 0.5% w/v yeast extract, 0.5% w/v NaCl) and cultivated overnight at 37°C. All entry and destination plasmids were validated by restriction digest and Sanger sequencing.

**Table 5.** Primers used for cloning the entry vectors.

| Name | Primer |
|------|--------|
| proBAM1 (pGD288) | F: AAAGGTCTCAACCTATGATCCGATCCTCAAAAGTATGTA<br>R: AAAGGTCTCATGTTTCTCTCTATCTCTCTTGTGTG |
| BAM1_CDS (pGD351) | F: TTTGGTCTCAGGCTCTATGAAACTTTTTCTTCTCCTTC<br>R:TTTGGTCTCACTGATAGATTGAGTAGATCCGGC<br>BsaI-site_#1_F: CTTGATCTCTCCGGACTCAACCTCTCCGG<br>BsaI-site_#1_R: CCGGAGAGGTTGAGTCCGGAGAGATCAAG<br>BsaI-site_#2_F: CTCATGTTGCTGACTTTGGACTCGCTAAATTCCTTCAAG<br>BsaI-site_#2_R: CTTGAAGGAATTTAGCGAGTCCAAAGTCAGCAACATGAG |
| proCLE40 | F: CACCGTTAAGCCAAGTAAGTACCACACAGC<br>R: CATTTCAAAAACCTCTTTGTG |
| proCLV1 | F: AAAGGTCTCAACCTGACTATTGTTTATACTTAGTTG<br>R: TTTGGTCTCATGTTCATTTTTTTAGTGTCCTC |
| CLV1_CDS | F: AAAGGTCTCAGGCTTAATGGCGATGAGAC<br>R: TTTGGTCTCACTGAACGCGATCAAGTTC<br>BasI-site_#1_F: CTAAAGGACACGGACTGCACGACTG<br>BasI-site_#1_R: CAGTCGTGCAGTCCGTGTCCTTTAG<br>BasI-site_#2_F: CTTAGAGTATCTTGGACTGAACGGAGCTGG<br>BasI-site_#2_R: CCAGCTCCGTTCAGTCCAAGATACTCTAAG |
| proCLV3 | F: AAAGGTCTCAACCTCGGATTATCCATAATAAAAAC<br>R:AAAGGTCTCATGTTTTTTAGAGAGAAAGTGACTGAG |
| tCLV3 | F: TTTGGTCTCTCTGCCGCCCTAATCTCTTGTT<br>R: TTTGGTCTCGTGATATGTGTGTTTTTTCTAAACAATC |

## Generation of stable *A. thaliana* lines

Generation of stable *A. thaliana* lines was done by using the floral dip method (*Clough and Bent, 1998*).

Translational *CLV1* (*CLV1:CLV1-GFP*) and transcriptional *CLV3* (*CLV3:NLS-3xmCherry*) reporter carry the BASTA plant resistance cassette. T1 seeds were sown on soil and sprayed with Basta (120 mg/mL) at 5 and 10 DAG. Seeds of ~10 independent Basta-resistant lines were harvested. The transcriptional CLE40 (*CLE40:Venus-HB*) reporter carries the hygromycin plant resistance cassette. T1 seeds were sown on ½ MS media containing 15 µg/mL hygromycin. The translational *BAM1* (*BAM1:BAM1-GFP*) reporter line carries a D-Alanin resistance cassette and T1 seeds were sown on ½ MS media containing 3–4 mM D-Alanin. Only viable plants (~10 T1 lines) were selected for the T2 generation. T2 seeds were then selected on ½ MS media supplied with either 15 µg/mL hygromycin, 3–4 mM D-Alanin or 10 µg/ mL of DL-phosphinothricin (PPT) as a BASTA alternative. Only plants from lines showing about ~75% viability were kept and cultivated under normal plant conditions (21°C, LD). Last, T3 seeds were plated on ½ MS media supplied with 3–4 mM D-Alanin or PPT again and plant lines showing 100% viability were kept as homozygous lines. The *CLE40:Venus-H2B*, *CLV3:NLS-3xmCherry* and *CLV1:CLV1-GFP* constructs were transformed into *Col-0* wild-type plants, and stable T3 lines were generated. Plants carrying the *CLE40:Venus-H2B* reporter were crossed into homozygous *clv3-9* or heterozygous *wus-7* mutants. Homozygous *clv3-9* mutants were detected by its obvious phenotype and were bred into a stable F3 generation. Homozygous *wus-7* mutants were identified by phenotype and DNA geno- type. Seeds were kept in the F2 generation since homozygous *wus-7* plants do not develop seeds. The *CLE40:Venus-H2B* reporter line was crossed with the *CLV3:NLS-3xmCherry* reporter line and was brought into a stable F3 generation. To generate the *CLE40:Venus-H2B//CLV3:WUS* line, plants carrying the *CLE40:Venus-H2B* line were transformed with the *CLV3:WUS* construct. T1 seeds were sown on 10 µg/mL of DL-phosphinothricin (PPT) and the viable seedlings were imaged. Plants carrying the *CLV1:CLV1-GFP* construct were crossed into *bam1-3*, *cle40-2*, *clv3-9* and *clv1-101* mutants until a homozygous mutant background was reached. *BAM1:BAM1-GFP* lines were transformed into *bam1-3* mutants and subsequently crossed into the *clv1-20* mutant background which rescued the extremely fasciated meristem phenotype of *bam1-3;clv1-20* double mutants (*Figure 5D–F*). *BAM1:BAM1-GFP// bam1-3* plants were also crossed into *cle40-2* and *clv3-9* mutants until a homozygous mutant back- ground was achieved. The *CLE40:CLE40-GFP* line was previously described in *Stahl et al., 2009*. The *WUS:NLS-GFP;CLV3:NLS-mCherry* reporter line was a gift from the Lohmann lab and was crossed into *clv3-9*, *cle40-2*, *clv1-101* and *bam1-3* mutants until a stable homozygous F3 generation was reached respectively.

Detailed information of all used *A. thaliana* lines can be found in *Table 6*.

## Confocal imaging of IFMs

To image IFMs in vivo, plants were grown under LD (16 hr light/8 hr dark) conditions and inflorescences were cut off at 5 or 6 WAG. Inflorescences were stuck on double-sided adhesive tape on an objective slide and dissected until only the meristem and primordia from P0 to maximum P10 were visible. Next, inflorescences were stained with either propidium iodide (PI 5 mM) or 4',6-diamidin-2-phenylindol (DAPI 1 µg/mL) for 2–5 min. Inflorescences were then washed three times with water and subsequently covered with water and a cover slide and placed under the microscope. Imaging was performed with a Zeiss LSM780 or LSM880 using a W Plan-Apochromat 40×/1.2 objective. Laser excitation, emission detection range and detector information for fluorophores and staining can be found in *Table 7*. All IFMs were imaged from the top taking XY images along the Z axis, resulting in a Z-stack through the inflorescence. The vegetative meristems were imaged as described for IFMs. Live imaging of the reporter lines in *A. thaliana* plants was performed by dissecting primary inflorescences (except for *clv3-9* mutants) at 5 WAG under LD conditions. For imaging of the reporter lines in the mutant back- grounds of *clv3-9*, secondary IFMs were dissected since the primary meristems are highly fasciated. Vegetative meristems were cultivated in continuous light conditions at 21°C on ½ MS media plates and were imaged at 10 DAG. For each reporter line, at least three independent experiments were performed and at least five IFMs were imaged.

**Table 6.** *Arabidopsis* lines that were analysed in this study.

| Name/construct | Background | Plant resistance | Generation | Reference |
|---|---|---|---|---|
| *BAM1:BAM1-GFP* | *bam1-3* | D-Ala | T4 | This study |
| *BAM1:BAM1-GFP* | *bam1-3;clv1-20* | D-Ala | F3 | This study |
| *BAM1:BAM1-GFP* | *bam1-3;clv3-9* | D-Ala | F3 | This study |
| *BAM1:BAM1-GFP* | *bam1-3;cle40-2* | D-Ala | F3 | This study |
| *CLE40:Venus-H2B* | *Col-0* | Hygromycin | T5 | This study |
| *CLE40:Venus-H2B* | *clv3-9* | Hygromycin | F3 | This study |
| *CLE40:Venus-H2B* | *wus-7* | Hygromycin | F2 | This study |
| *CLE40:Venus-H2B* | *CLV3:WUS//Col-0* | Hygromycin/ Basta | T1* | This study |
| *CLE40:CLE40-GFP* | *Col-0* | N/A | T3 | **Stahl et al., 2009** |
| *CLV1:CLV1-GFP* | *Col-0* | Basta | T4 | This study |
| *CLV1:CLV1-GFP* | *bam1-3* | Basta | F3 | This study |
| *CLV1:CLV1-GFP* | *clv3-9* | Basta | F3 | This study |
| *CLV1:CLV1-GFP* | *cle40-2* | Basta | F3 | This study |
| *CLV1:CLV1-2xGFP* | *clv1-11* | Basta | N/A | **Nimchuk et al., 2011** |
| *CLV3:NLS-3xmCherry* | *CLE40:Venus-H2B//Col-0* | Basta/ hygromycin | F3 | This study |
| *CLV3:NLS-mCherry WUS:NLS-GFP* | *Col-0* | Kanamycin | N/A | Anne Pfeiffer |
| *CLV3:NLS-mCherry WUS:NLS-GFP* | *cle40-2* | Kanamycin | F3 | This study |
| *CLV3:NLS-mCherry WUS:NLS-GFP* | *bam1-3* | Kanamycin | F3 | This study |
| *CLV3:NLS-mCherry WUS:NLS-GFP* | *clv1-101* | Kanamycin | F3 | This study |
| *CLV3:NLS-mCherry WUS:NLS-GFP* | *clv3-9* | Kanamycin | F3 | This study |

*Plants do not overcome seedling stage.

**Table 7.** Microscopy settings used for imaging.

| Fluorophore/ staining | Excitation (nm) | Emission (nm) | MBS | Detector | Light source |
|---|---|---|---|---|---|
| DAPI | 405 | 410–490 | 405 | PMT | Diode |
| GFP | 488 | 500–545 | 488/561 | GaAsP | Argon laser |
| Venus | 514 | 518–540 | 458/514 | GaAsP | Argon laser |
| mCherry | 561 | 570–640 | 458/561 | PMT | DPSS laser |
| PI | 561 | 595–650 | 488/561 | PMT | DPSS laser |

PMT, photomultiplier tubes; DPSS, diode-pumped solid state.

## Phenotyping of *CLV* mutants

For meristem measurements (area size, width and height), primary and secondary IFMs of wild-type (*Col-0*) and mutant plants (*cle40-2, cle40-cr1-3, bam1-3, cle40-2;bam1-3, clv1-101*) were dissected at 6 WAG under LD conditions. For *clv3-9* and *clv1-101;bam1-3*, only secondary IFMs were imaged and analysed due to the highly fasciated primary meristems. Longitudinal optical sections of the Z-stacks were performed through the middle of the meristem starting in the centre of primordia P5 and ending in the centre of primordia P4. Based on the longitudinal optical sections (XZ), meristem height and area size were measured as indicated in *Figure 6*. IFM sizes from *Figure 1E* are also used in *Figure 6E* for *Col-0*, *cle40-2* and *clv3-9* plants.

Same procedure was used to count the cells expressing *WUS* in different mutant backgrounds (*Figure 7A–E*). Longitudinal optical sections of IFMs at 5 WAG were performed from P4 to P5, and only nuclei within the meristem area were counted and plotted. For analyses of carpel numbers, the oldest 10–15 siliques per plant at 5 WAG were used. Each carpel was counted as 1, independent of its size. N number depicts number of siliques. Leaf measurements were performed at 4 WAG, and four leaves of each plant were measured and plotted. Data was obtained from at least three independent experiments.

## Root length assay

Effects of CLE40 peptide treatment on root growth were analysed by cultivating seedlings (*Col-0*, *clv1-101*, *bam1-3* and *bam1-3;clv1-101*) on ½ MS agar plates (squared) supplied with or without synthetic CLE40p at indicated concentrations. The plates were kept upright in continuous light at 21°C and 60% relative humidity. Root growth was measured at 11 DAG by scanning plates and analysed using ImageJ to measure root lengths. For each genotype and condition, 20–48 single roots were measured. Graphs and statistical analyses were done with Prism v.8.

## Data analysis

For visualization of images, the open-source software ImageJ v 1.53c (*Schneider et al., 2012*) was used. All images were adjusted in 'Brightness and Contrast'. IFMs in *Figure 7* were imaged with identical microscopy settings (except for *clv3-9* mutants) and were all changed in 'Brightness and Contrast' with the same parameters to ensure comparability. *clv3-9* mutants were imaged with a higher laser power since meristems are highly fasciated. MIPs were created by using the 'Z-Projection' function and longitudinal optical sections were performed with the 'Reslice…' function, resulting in the XZ view of the image. Meristem width, height and area size were measured with the 'Straight line' for width and height and the 'Polygon selection' for area size. The shape parameter σ was calculated by the quotient of height and width from each IFM. For L1 visualization, the open-source software MorphoGraphX (https://www.mpipz.mpg.de/MorphoGraphX/) was used that was developed by Richard Smith. 2½ D images were created by following the steps in the MorphoGraphX manual (*Barbier de Reuille et al., 2015*). After both channels (PI and fluorophore signal) were projected to the created mesh, both images were merged using ImageJ v 1.53c.

For all statistical analyses, GraphPad Prism v8.0.0.224 was used. Statistical groups were assigned after calculating p-values by ANOVA and Tukey's or Dunnett's multiple comparison test (differential grouping from p≤0.01) as indicated under each figure. Same letters indicate no statistical differences.

Intensity plot profiles were measured with the 'plot profile' function in Fiji and plotted in GraphPad Prism. Each intensity profile was normalized. For each genotype, nine meristems were analysed and the mean of all nine meristems with its corresponding error bars (standard deviation) was plotted (*Figure 5—figure supplement 3*).

Imaris software was used to detect *WUS*-expressing cells within the entire IFMs (MIP) of different mutant backgrounds. The 'spot detection' function in Imaris was used with the same algorithm for *Col-0*, *cle40-2*, *bam1-3* genotypes ([Algorithm] Enable Region Of Interest = false; Enable Region Growing = false; Enable Tracking = false; [Source Channel]; Source Channel Index = 2; Estimated; Diameter = 3.00 um; Background Subtraction = true; [Classify Spots] "Quality" above 3000). Due to the highly fasciated meristems in *clv1-101* and *clv3-9* mutants, the threshold for 'Quality' for *WUS*-expressing cells was set to 1000.

All plasmid maps and cloning strategies were created and planned using the software VectorNTI.

## Acknowledgements

This study was funded by DFG through iGrad-Plant (IRTG 2466), CRC 1208 and CEPLAS (EXC 2048). We thank Cornelia Gieseler, Silke Winters and Carin Theres for technical support and Yasuka L Yamaguchi (Sawa lab) and Anne Pfeiffer (Lohmann lab) for sharing *Arabidopsis* seeds. We also thank Vicky Howe for proofreading the manuscript, the Center for Advanced imaging (CAi) at HHU for microscopy support and Aleksandra Sapala for support with MorphoGraphX.

## Additional information

### Funding

| Funder | Grant reference number | Author |
| --- | --- | --- |
| Deutsche Forschungsgemeinschaft | CEPLAS (EXC2048) | Rüdiger GW Simon |
| Deutsche Forschungsgemeinschaft | iGRAD-PLANT | Rüdiger GW Simon Rene Wink Jenia Schlegel |
| Deutsche Forschungsgemeinschaft | CRC1208 | Rüdiger GW Simon Gregoire Denay |

The funders had no role in study design, data collection and interpretation, or the decision to submit the work for publication.

### Author contributions

Jenia Schlegel, Investigation, Writing - original draft, Writing - review and editing; Gregoire Denay, Karine Gustavo Pinto, Yvonne Stahl, Julia Schmid, Patrick Blümke, Investigation; Rene Wink, Resources; Rüdiger GW Simon, Conceptualization, Funding acquisition, Investigation, Project administration, Supervision, Writing - original draft, Writing - review and editing

### Author ORCIDs

Jenia Schlegel  http://orcid.org/0000-0003-0434-4479
Gregoire Denay  http://orcid.org/0000-0002-8850-3029
Patrick Blümke  http://orcid.org/0000-0001-7315-6792
Rüdiger GW Simon  http://orcid.org/0000-0002-1317-7716

### Decision letter and Author response

Decision letter https://doi.org/10.7554/eLife.70934.sa1
Author response https://doi.org/10.7554/eLife.70934.sa2

## Additional files

### Supplementary files

• Transparent reporting form

### Data availability

Original microscopy and image analysis data represented in the manuscript are available via Dryad (https://doi.org/10.5061/dryad.1g1jwstwf) and BioStudies (https://www.ebi.ac.uk/biostudies/studies/S-BSST723).

The following dataset was generated:

| Author(s) | Year | Dataset title | Dataset URL | Database and Identifier |
|---|---|---|---|---|
| Schlegel J, Denay G, Stahl Y, Schmid J, Blümke P H, Wink R G, Pinto K, Simon R | 2021 | Control of Arabidopsis shoot stem cell homeostasis by two antagonistic CLE peptide signalling pathways | https://www.ebi.ac.uk/biostudies/studies/S-BSST723 | Dryad Digital Repository, 10.5061/dryad.1g1jwstwf |
| Schlegel J | 2021 | Control of Arabidopsis shoot stem cell homeostasis by two antagonistic CLE peptide signalling pathways | https://www.ebi.ac.uk/biostudies/studies/S-BSST723 | BioStudies, S-BSST723 |

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

# Appendix 1

### Appendix 1—key resources table

| Reagent type (species) or resource | Designation | Source or reference | Identifiers | Additional information |
|---|---|---|---|---|
| Strain, strain background (*Arabidopsis thaliana*) | Columbia (*Col-0*) | NASC ID: N22625 | ABRC: CS22625 | |
| Genetic reagent (*A. thaliana*) | *bam1-3* | **Alonso et al., 2003**; NASC ID: N515302 | ABRC: SALK_015302 | T-DNA mutation in *Col-0* background |
| Genetic reagent (*A. thaliana*) | *cle40-2* | **Stahl et al., 2009** | NA | Transposon mutation *Col-0* background |
| Genetic reagent (*A. thaliana*) | *cle40-cr1/2/3* | **Yamaguchi et al., 2017** | NA | CRISPR in *Col-0* background |
| Genetic reagent (*A. thaliana*) | *clv3-9* | **Hobe et al., 2003** | NA | EMS in *Col-0* background |
| Genetic reagent (*A. thaliana*) | *clv1-20* | **Alonso et al., 2003**; NASC ID: N508670 | ABRC: SALK_008670 | T-DNA mutation in *Col-0* background |
| Genetic reagent (*A. thaliana*) | *clv1-101* | **Alonso et al., 2003**; NASC ID: N858348 | ABRC: CS858348 | T-DNA mutation in *Col-0* background |
| Genetic reagent (*A. thaliana*) | *wus-7* | **Graf et al., 2010**; | NA | EMS in L.*er.* background Gift from J. Lohmann lab |
| Genetic reagent (*A. thaliana*) | *BAM1:BAM1-GFP* | This study | NA | Transgenic line in *bam1-3* background |
| Genetic reagent (*A. thaliana*) | *BAM1:BAM1-GFP* | This study | NA | Transgenic line in *bam1-3;clv1-20* background |
| Genetic reagent (*A. thaliana*) | *BAM1:BAM1-GFP* | This study | NA | Transgenic line in *bam1-3;clv3-9* background |
| Genetic reagent (*A. thaliana*) | *BAM1:BAM1-GFP* | This study | NA | Transgenic line in *bam1-3;cle40-2* background |
| Genetic reagent (*A. thaliana*) | *CLE40:Venus-H2B* | This study | NA | Transgenic line in *Col-0* background |
| Genetic reagent (*A. thaliana*) | *CLE40:Venus-H2B* | This study | NA | Transgenic line in *clv3-9* background |
| Genetic reagent (*A. thaliana*) | *CLE40:Venus-H2B* | This study | NA | Transgenic line in *wus-7* background |
| Genetic reagent (*A. thaliana*) | *CLE40:Venus-H2B* | This study | NA | Transgenic line in *CLV3:WUS/Col-0* background |
| Genetic reagent (*A. thaliana*) | *CLE40:CLE40-GFP* | **Stahl et al., 2009** | NA | Transgenic line in *Col-0* background |
| Genetic reagent (*A. thaliana*) | *CLV1:CLV1-GFP* | This study | NA | Transgenic line in *Col-0* background |
| Genetic reagent (*A. thaliana*) | *CLV1:CLV1-GFP* | This study | NA | Transgenic line in *clv1-101* background |
| Genetic reagent (*A. thaliana*) | *CLV1:CLV1-GFP* | This study | NA | Transgenic line in *bam1-3* background |
| Genetic reagent (*A. thaliana*) | *CLV1:CLV1-GFP* | This study | NA | Transgenic line in *clv3-9* background |
| Genetic reagent (*A. thaliana*) | *CLV1:CLV1-GFP* | This study | NA | Transgenic line in *cle40-2* background |
| Genetic reagent (*A. thaliana*) | *CLV1:CLV1-2xGFP* | **Nimchuk et al., 2011** | NA | Transgenic line in *clv1-11* background Gift from Z. Nimchuk lab |
| Genetic reagent (*A. thaliana*) | *CLV3:NLS-3xmCherry* | This study | NA | Transgenic line in *CLE40:Venus-H2B/Col-0* background |

*Appendix 1 Continued on next page*

*Appendix 1 Continued*

| Reagent type (species) or resource | Designation | Source or reference | Identifiers | Additional information |
| --- | --- | --- | --- | --- |
| Genetic reagent (*A. thaliana*) | *CLV3:NLS-mCherry WUS:NLS-GFP* | Anne Pfeiffer | NA | Transgenic line in *Col-0* background Gift from J. Lohmann lab |
| Genetic reagent (*A. thaliana*) | *CLV3:NLS-mCherry WUS:NLS-GFP* | This study | NA | Transgenic line in *cle40-2* background |
| Genetic reagent (*A. thaliana*) | *CLV3:NLS-mCherry WUS:NLS-GFP* | This study | NA | Transgenic line in *bam1-3* background |
| Genetic reagent (*A. thaliana*) | *CLV3:NLS-mCherry WUS:NLS-GFP* | This study | NA | Transgenic line in *clv1-101* background |
| Genetic reagent (*A. thaliana*) | *CLV3:NLS-mCherry WUS:NLS-GFP* | This study | NA | Transgenic line in *clv3-9* background |
| Strain, strain background (*Agrobacterium tumefaciens*) | *A. tumefaciens* GV3101 pMP90 pSoup | Lifeasible | Cat# ACC-101 | |
| Recombinant DNA reagent | pGGZ001 | *Lampropoulos et al., 2013* | Addgene | RRID:Addgene_48868 |
| Recombinant DNA reagent | pGGB002 | *Lampropoulos et al., 2013* | Addgene | RRID:Addgene_48820 |
| Recombinant DNA reagent | pGGE009 | *Lampropoulos et al., 2013* | Addgene | RRID:Addgene_48841 |
| Recombinant DNA reagent | pGGF003 | *Lampropoulos et al., 2013* | Addgene | RRID:Addgene_48844 |
| Recombinant DNA reagent | pGGC026 | *Lampropoulos et al., 2013* | Addgene | RRID:Addgene_48831 |
| Recombinant DNA reagent | pGGD002 | *Lampropoulos et al., 2013* | Addgene | RRID:Addgene_48834 |
| Recombinant DNA reagent | pGGF008 | *Lampropoulos et al., 2013* | Addgene | RRID:Addgene_48848 |
| Recombinant DNA reagent | pMDC99 | *Curtis and Grossniklaus, 2003* | NA | |
| Recombinant DNA reagent | pBAM1/pGGA000 | This study | NA | Entry vector used for cloning See Tables 4 and 5 for details |
| Recombinant DNA reagent | BAM1-CDS/pGGC000 | This study | TAIR: AT5G65700 | Entry vector used for cloning See Tables 4 and 5 for details |
| Recombinant DNA reagent | pCLV1-/pGGA000 | This study | NA | Entry vector used for cloning See Tables 4 and 5 for details |
| Recombinant DNA reagent | CLV1-CDS/pGGC000 | This study | TAIR: AT1G75820 | Entry vector used for cloning See Tables 4 and 5 for details |
| Recombinant DNA reagent | pCLE40/pGGA000 | This study | NA | Entry vector used for cloning See Tables 4 and 5 for details |
| Recombinant DNA reagent | pCLV3/pGGA000 | This study | NA | Entry vector used for cloning See Tables 4 and 5 for details |
| Recombinant DNA reagent | pCLV3/pGGE000 | This study | NA | Entry vector used for cloning See Tables 4 and 5 for details |
| Sequence-based reagent | Cloning primers | This study | NA | *Table 5* |
| Sequence-based reagent | Genotyping primers | This study | NA | *Table 2* |
| Chemical compound, drug | BASTA non-selective herbicide | Bayer CropScience | 84442615 | |
| Peptide, recombinant protein | Synthetic CLE40 | Peptides&Elephants | NA | |

*Appendix 1 Continued on next page*

*Appendix 1 Continued*

| Reagent type (species) or resource | Designation | Source or reference | Identifiers | Additional information |
|---|---|---|---|---|
| Software, algorithm | ImageJ v 1.53c | *Schneider et al., 2012* | https://imagej.net/software/fiji/ | RRID:SCR_003070 |
| Software, algorithm | MorphoGraphX | *Barbier de Reuille et al., 2015* | https://www.mpipz.mpg.de/MorphoGraphX/ | |
| Software, algorithm | GraphPad Prism v8.0.0.224 | NA | GraphPad Prism (https://graphpad.com) | RRID:SCR_002798 |
| Software, algorithm | Imaris | NA | http://www.bitplane.com/imaris/imaris | RRID:SCR_007370 |

