## [Decision Letter]

**Acceptance summary:**

We congratulate the authors for this outstanding, mechanistic work on stem cell regulation in plants. This study provides a breakthrough in understanding the homeostasis of plant stem cells by providing a mechanistic view on the CLE40-BAM1-WUS pathway. The authors thereby provide a concise model on the regulation of the shoot apical meristem.

**Decision letter after peer review:**

Thank you for submitting your article "Control of Arabidopsis shoot stem cell homeostasis by two antagonistic CLE peptide signalling pathways" for consideration by *eLife*. Your article has been reviewed by 3 peer reviewers, including Sheila McCormick as Reviewing Editor and Reviewer #1, and the evaluation has been overseen by Jürgen Kleine-Vehn as the Senior Editor.

Essential revisions:

1) Please address the issues pointed out in the public reviews, most importantly, the validation of reporters and quantification of imaging results across multiple samples.

2) Additional data on the genetic interactions of BAM1 and CLV1 would also be welcome, if at all possible, since the key question of antagonism vs. redundancy is difficult to answer confidently without this knowledge.

*Reviewer #1 (Recommendations for the authors):*

Lines 122-125 are a bit confusing, as they say functions for CLE40 have not previously been described, but then state that overexpression causes shoot stem cell termination. – I think it would be better to say ENDOGENOUS functions for CLE40 have not previously been described, ALTHOUGH overexpression……, i.e. that the overexpression phenotype is due to the functional overlap of CLE40 function with CLV3 peptide function, and does not say anything about the role of CLE40.

Lines 146-150. Why discuss the presence of introns, introns have nothing to do with protein function, and anyway, the functional part of any CLE protein is the cleaved dodecapeptide.

I was surprised that the last paragraph of the discussion talks about what might be happening in monocots. I think this is a weak ending, since they did nothing with monocots. I would end the discussion at line 560.

*Reviewer #2 (Recommendations for the authors):*

Suggestions for improvement

Introduction:

I found it a bit odd that the bryophyte text came before the grasses- re-order to consider the angiosperm node and then land plant node of the plant tree of life?

Results:

Figure 1:

Figure 1A doesn't show phylogenetic clustering as discussed in the text. Either show clustering or change the text to peptide structure and include some peptides from the other species discussed. I really liked the apical dome images shown in Fig6Suppl.Figure 2, and something like this would have been very helpful as part of Figure 1 and other figures with the bee swarm plots of IFM area as a quick visual readout of the shape differences. In the text relating to Figure 1, percentage changes are given, but in the graph absolute values are shown. It would have been helpful to see the percentage changes.

Figure 2:

It would be very helpful to identify the organs referred to in the text with arrowheads. In panel C, expression looks downregulated rather than absent. How was the CZ defined in panel B?

Figure 3:

It would be very helpful to identify the organs referred to in the text with arrowheads.

Figure 4:

The localization data look great. Please explain why new CLV1 translational fusions were generated and how the results obtained compare to work published by Nimchuk et al., in 2011. How does expression in primordia fit with previous work?

Figure 5:

Highlight the OC?

Figure 6:

It is hard to see the labels in A and B- box behind as for C and D? Panel E- show % change data? Show meristem shape shapshots in graph?

Figure 7

Show meristem shape shapshots in graph?

Figure 8

This is slightly unclear- are the black bound connections between cells plasmodesmata? If so clarify.

How do the model and ideas about diffusion on line 536 fit with ideas about symplastically isolated domains in shoot apices cf. Gisel et al., (1999) in Development and others?

*Reviewer #3 (Recommendations for the authors):*

Before acceptance by *eLife* the authors need to address the issues pointed out in the public review, most importantly, validation of reporters and quantification of imaging results across multiple samples. Experiments on making BAM1 expression independent from CLV1 activity would be important, but not essential.

Additional points are:

The reference for the CLE40 reporter construct (Wink, R., 2013, most likely a PhD thesis) does not appear to be public. This needs to be fixed.

In the attempt to retrieve the above obscure reference, a PhD thesis from the Simon lab was identified: (Schmid, J., "On the Role of CLE40 A Peptide with Antagonistic Functions in *Arabidopsis thaliana* Shoot Meristem Development", 2015). Worryingly, images in this thesis show an apparently different expression pattern for a CLE40 reporter, which largely overlaps with the WUS expression domain in the OC (Figure 6, A-C). Since the thesis also cites Wink, R. 2013 for the reporter, one has to assume that we are looking at the same plant line with two opposing expression patterns. One of the chapters of the thesis even reads"CLE40 and CLV1 have largely overlapping expression domains". The authors can probably explain the discrepancies, but it just underlines the importance of a proper validation of the used experimental tools in the manuscript.

In Figure 4 and 5 the signal of the CLV1 and BAM1 reporters are obscured by the π stain. It would be better to show the reporter channels separately. The images in Figure 6 are largely redundant with the images presented in Figures 2,4 and 5 and do not offer any additional information. Instead Figures 2,4 and 5 could easily be condensed to one figure without a significant loss of information.

The term optical section is very often used in a confusing way throughout the manuscript. Maybe the authors could rather refer to the geometric plane they chose to show from an image volume, e.g. transversal or longitudinal.

At least two different clv1 alleles were used. Please provide information about the different alleles and why they were used.

In Figure 7G, there is no need to split the y-axis. This should be corrected.

Figure 1-SupplFigure 1 is dispensable since the leaf length appears to be largely irrelevant for the manuscript.

In Fig8-SupplFigure 1-4: If no picture of CLE40 was acquired there is no reason to show a respective image panel.

---

## [Author Response]

Reviewer #1 (Recommendations for the authors):Lines 122-125 are a bit confusing, as they say functions for CLE40 have not previously been described, but then state that overexpression causes shoot stem cell termination. – I think it would be better to say ENDOGENOUS functions for CLE40 have not previously been described, ALTHOUGH overexpression……, i.e. that the overexpression phenotype is due to the functional overlap of CLE40 function with CLV3 peptide function, and does not say anything about the role of CLE40.

We followed the reviewers suggestion.

Lines 146-150. Why discuss the presence of introns, introns have nothing to do with protein function, and anyway, the functional part of any CLE protein is the cleaved dodecapeptide.

Introns can inform us on the evolutionary relationship between genes, we would therefore suggest to keep this information in the manuscript. We added an explanatory sentence to support this notion.

I was surprised that the last paragraph of the discussion talks about what might be happening in monocots. I think this is a weak ending, since they did nothing with monocots. I would end the discussion at line 560.

Research on stem cell control pathways in monocots has been in focus recently, and we believe that a comparison between dicots and monocots is valuable and will be of benefit for the reader to understand the significance of our findings.

Reviewer #2 (Recommendations for the authors):Suggestions for improvementIntroduction:I found it a bit odd that the bryophyte text came before the grasses- re-order to consider the angiosperm node and then land plant node of the plant tree of life?

Several recent studies reported on CLV-like pathways in bryophytes, and in light of such recent discoveries, we started with a discussion of these results.

Results:Figure 1:Figure 1A doesn't show phylogenetic clustering as discussed in the text. Either show clustering or change the text to peptide structure and include some peptides from the other species discussed.

We changed the text according to your recommendations.

I really liked the apical dome images shown in Fig6Suppl.Figure 2, and something like this would have been very helpful as part of Figure 1 and other figures with the bee swarm plots of IFM area as a quick visual readout of the shape differences.

Thank you for the recommendation, we redesigned Fig6 accordingly.

In the text relating to Figure 1, percentage changes are given, but in the graph absolute values are shown. It would have been helpful to see the percentage changes.

We added percentage changes to Figure 1.

Figure 2:It would be very helpful to identify the organs referred to in the text with arrowheads.

We labelled organs in the Figure 2

In panel C, expression looks downregulated rather than absent.

We corrected the description for panel C in the text and legend.

How was the CZ defined in panel B?

We eliminated the marking of the CZ, since there is no independent parameter that we could use to unequivocally identify only the CZ.

Figure 3:It would be very helpful to identify the organs referred to in the text with arrowheads.

Organs are now labelled.

Figure 4:The localization data look great.

Thank you.

Please explain why new CLV1 translational fusions were generated and how the results obtained compare to work published by Nimchuk et al., in 2011. How does expression in primordia fit with previous work?

We now added a new Figure 4-Suppl Figure 2, which serves to compare the expression patterns of the different transgenic lines, which are actually identical in terms of expression pattern. Our reasoning for creating new transgenic lines was that we wanted to exclude artefacts due to vector backbones, or to the fusion of two GFPs to CLV1 in case of the lines generated by Zack Nimchuk. We dare to say that our expression analysis using both the Nimchuk lines and our lines goes further into detail than previous studies, and we therefore were able to detect and describe the CLV1 expression profile with much higher spatial resolution than before.

Figure 5:Highlight the OC?

We highlighted the OC in Figure 5.

Figure 6:It is hard to see the labels in A and B- box behind as for C and D?

We changed Figure 6, and the requested labels and boxes are now to be found in Figure 5-Suppl Figure 5.

Panel E- show % change data? Show meristem shape shapshots in graph?

Adding % change might result in an "overloading" of the graph, but we added meristem shape snapshots to the new Figure 6.

Figure 7Show meristem shape shapshots in graph?

Meristem shapes are shown in the figure parts A´ to F´.

Figure 8This is slightly unclear- are the black bound connections between cells plasmodesmata? If so clarify.

Yes, this shall indicate cytoplasmic connectivity between cells, mediated by plasmodesmata. We added a description to the legend.

How do the model and ideas about diffusion on line 536 fit with ideas about symplastically isolated domains in shoot apices cf. Gisel et al., (1999) in Development and others?

Symplastic domains are an interesting concept, but have not been understood from a mechanical or biochemical point of view. Symplastic isolation was shown to affect the mobility of proteins and also (more recently) for specific miRNAs, under some circumstances. We here postulate that a molecule X mediates communication between domains. If this is a small molecule, such as a phytohormone, it might not be restricted by symplastic traffic control.

Reviewer #3 (Recommendations for the authors):Before acceptance by eLife the authors need to address the issues pointed out in the public review, most importantly, validation of reporters and quantification of imaging results across multiple samples.

We have now added multiple data sets validating reporter lines and presenting more of the imaging results that we obtained from multiple samples.

See new Supplementary Figure 2-Suppl Figure 2, Figure 4-Suppl Figure 1, Figure 4-Suppl Figure 2, Figure 4-Suppl Figure 3, Figure 5-Suppl Figure 1, Figure 5-Suppl Figure 2, Figure 5-Suppl Figure 3, Figure 5-Suppl Figure 4

Experiments on making BAM1 expression independent from CLV1 activity would be important, but not essential.Additional points are:The reference for the CLE40 reporter construct (Wink, R., 2013, most likely a PhD thesis) does not appear to be public. This needs to be fixed.

Our apologies for that omission! We now describe the CLE40 reporter design in Materials and methods. An earlier description of a CLE40 reporter can also be found already in Stahl et al., 2009.

In the attempt to retrieve the above obscure reference, a PhD thesis from the Simon lab was identified: (Schmid, J., "On the Role of CLE40 A Peptide with Antagonistic Functions in *Arabidopsis thaliana* Shoot Meristem Development", 2015). Worryingly, images in this thesis show an apparently different expression pattern for a CLE40 reporter, which largely overlaps with the WUS expression domain in the OC (Figure 6, A-C). Since the thesis also cites Wink, R. 2013 for the reporter, one has to assume that we are looking at the same plant line with two opposing expression patterns. One of the chapters of the thesis even reads"CLE40 and CLV1 have largely overlapping expression domains". The authors can probably explain the discrepancies, but it just underlines the importance of a proper validation of the used experimental tools in the manuscript.

Our PhD student Julia Schmid did not image through the meristem centre, and therefore missed the lack of CLE40 expression in that region. We can illustrate this with (Author response image 1).

**Author response image 1. sa2fig1:** 

In Figure 4 and 5 the signal of the CLV1 and BAM1 reporters are obscured by the π stain. It would be better to show the reporter channels separately.

We now show the channels separately.

The images in Figure 6 are largely redundant with the images presented in Figures 2,4 and 5 and do not offer any additional information. Instead Figures 2,4 and 5 could easily be condensed to one figure without a significant loss of information.

The images in Figure 6 are now part of a supplemental figure. We would prefer to keep the other figures since only then can the reader obtain a full comprehensive view on the expression pattern.

See new Figure 6 and Figure 5-Suppl Figure 5

The term optical section is very often used in a confusing way throughout the manuscript. Maybe the authors could rather refer to the geometric plane they chose to show from an image volume, e.g. transversal or longitudinal.

We checked the manuscript and clarified the geometric planes of sections.

At least two different clv1 alleles were used. Please provide information about the different alleles and why they were used.

For each mutant genotype we always used at least two different alleles. The *clv1-20* mutant depicts a weak allele and results in a reduced mRNA level, while the *clv1-101* mutant has been described as a null mutant before ([41] ). We therefore used for most of our mutant studies the *clv1‑101* allele. The *clv1-20* mutant was only used in combination with the *bam1-3* mutant (in the rescue experiment), since both double mutants (*clv1‑20;bam1-*3 and *clv1-101;bam1-3*) show the same highly faceted phenotype (which could be rescued by the *BAM1:BAM1-GFP* reporter).

In Figure 7G, there is no need to split the y-axis. This should be corrected.

This was corrected.

Figure 1-SupplFigure 1 is dispensable since the leaf length appears to be largely irrelevant for the manuscript.

Leaf lengths has been used by other authors as an indicator for a loss of BAM1 function (DeYoung et al., 2006), so its one aspect of the *bam1* phenotype. We would therefore prefer to show these data, even if their significance in the context of meristem function is not yet apparent.

In Fig8-SupplFigure 1-4: If no picture of CLE40 was acquired there is no reason to show a respective image panel.

We adjusted the Figure 8-SupplFigure 1-4.